# ADP binding by the *Culex quinquefasciatus* mosquito D7 salivary protein enhances blood feeding on mammals

Ines Martin-Martin [1], Andrew Paige [1], Paola Carolina Valenzuela Leon [1], Apostolos G. Gittis[1], Olivia Kern[1], Brian Bonilla[1], Andrezza Campos Chagas[1], Sundar Ganesan[1], Leticia Barion Smith[1], David N. Garboczi[1] & Eric Calvo [1]✉

During blood-feeding, mosquito saliva is injected into the skin to facilitate blood meal acquisition. D7 proteins are among the most abundant components of the mosquito saliva. Here we report the ligand binding specificity and physiological relevance of two D7 long proteins from *Culex quinquefasciatus* mosquito, the vector of filaria parasites or West Nile viruses. CxD7L2 binds biogenic amines and eicosanoids. CxD7L1 exhibits high affinity for ADP and ATP, a binding capacity not reported in any D7. We solve the crystal structure of CxD7L1 in complex with ADP to 1.97 Å resolution. The binding pocket lies between the two protein domains, whereas all known D7s bind ligands either within the N- or the C-terminal domains. We demonstrate that these proteins inhibit hemostasis in ex vivo and in vivo experiments. Our results suggest that the ADP-binding function acquired by CxD7L1 evolved to enhance blood-feeding in mammals, where ADP plays a key role in platelet aggregation.

[1] Laboratory of Malaria and Vector Research, National Institute of Allergy and Infectious Diseases, National Institutes of Health, Rockville, MD 20852, USA.
✉email: ecalvo@niaid.nih.gov

**C**ulex quinquefasciatus (Diptera: Culicidae), commonly known as the southern house mosquito, is a vector of medical and veterinary importance of filaria parasites, including Wuchereria bancrofti and Dirofilaria immitis[1,2] and avian malaria parasites (Plasmodium relictum)[3]. It also can transmit several arboviruses including Rift Valley fever, West Nile, St. Louis or Western equine encephalitis viruses[4,5]. Adult female mosquitoes need to acquire vertebrate blood for egg development. During blood feeding, mosquito saliva is injected at the bite site and facilitates blood meal acquisition through anti-hemostatic compounds that prevent blood clotting, platelet aggregation, and vasoconstriction as well as host immune responses[6].

D7 proteins are among the most abundant components in the salivary glands of several blood-feeding arthropods and are distantly related to the arthropod odorant-binding protein super-family[7–10]. The D7s belong to a multi-gene family that evolved through gene duplication events, resulting in long forms and truncated versions of a duplicated long form, known as short forms[8]. In addition to gene duplication, D7 proteins have undergone functional divergence, resulting in binding specialization with different affinities for host biogenic amines, as seen in Anopheles gambiae D7 short forms[10]. The D7 proteins act as kratagonists, binding and trapping agonists of hemostasis, including biogenic amines and leukotrienes (LT)[8,11,12]. The D7 long protein from Anopheles stephensi and intermediate D7 forms from the sand fly Phlebotomus papatasi have lost the capacity to bind biogenic amines but have evolved the capability to scavenge thromboxane A2 (TXA$_2$) and LT[13,14], mediators of platelet aggregation and inflammation. Interestingly, an Aedes aegypti D7 long protein has a multifunctional mechanism of ligand binding: The N-terminal domain binds cysteinyl LT while the C-terminal domain shows high affinity to biogenic amines such as nor-epinephrine, serotonin, or histamine[10,11]. Many authors have studied this group of proteins since the first description of a D7 salivary protein in a blood-feeding arthropod[15–19]. Although the function of several mosquito D7 proteins including An. gambiae D7 short forms as well as the Ae. aegypti and An. ste-phensi long forms have been deciphered[10,11,13], the role of C. quinquefasciatus D7 proteins remains unknown.

In this work, we express, purify, and biochemically characterize the two D7 long forms, L1 and L2, from C. quinquefasciatus salivary glands. We show the different affinities for biogenic amines and eicosanoids to CxD7L2 and discover a function for CxD7L1. CxD7L1 binds adenosine 5′-monophosphate (AMP), adenosine 5′-diphosphate (ADP), adenosine 5′-triphosphate (ATP), and adenosine, which are essential agonists of platelet aggregation and act as inflammatory mediators. CxD7L1 shows no binding to biogenic amines or eicosanoids, that are previously described ligands for other D7 proteins[10,11,13]. We determine the crystal structure of CxD7L1 in complex with ADP and observe that the ADP binding pocket is located between the N-terminal and C-terminal domains. We also show that CxD7L1 and CxD7L2 act as platelet aggregation inhibitors ex vivo and inter-fere with blood hemostasis in vivo supporting the hypothesis that the binding of ADP by CxD7L1 helped C. quinquefasciatus to evolve from blood feeding on birds, where serotonin plays a key role in aggregation, to blood feeding on mammals where ADP is a key mediator of platelet aggregation.

## Results

### Characterization of CxD7L1 and CxD7L2. In previous studies[7,8], C. quinquefasciatus salivary gland cDNA libraries were sequenced resulting in the identification of 14 cDNA clusters with high sequence similarity to the previously known D7 long forms

(D7clu1: AF420269 and D7clu12: AF420270) and a D7 short form (D7Clu32, AF420271). We compared the amino acid sequence of C. quinquefasciatus D7 long proteins with other well-characterized D7 members, whose function and structure have been solved. Exonic regions were conserved for all previously studied mosquito proteins (Culex, Aedes, and Anopheles) and phlebotomine sand flies where the first exon corresponds to a secretion signal peptide and the mature proteins are encoded by exons 2, 3, 4, and 5 (Supplementary Fig. 1).

We named C. quinquefasciatus salivary long D7 proteins CxD7L1 (AAL16046) and CxD7L2 (AAL16047) and character-ized them by gene expression analysis and immunolocalization. To determine the stage, sex, and tissue specificity of the D7 protein transcripts, qPCR experiments were performed on all four larval instars, pupae, whole male, whole female, female head and thorax, and female abdomen. We confirmed that both transcripts are only found in female adult stages with similar levels of expression and specifically located in the head and thorax of the mosquito, where the salivary glands are located. No amplification of CxD7L1 and CxD7L2 transcripts was found in the abdomen (Fig. 1a). These results confirmed that CxD7L1 and CxD7L2 expression is unique to the female salivary glands of C. quinquefasciatus, as previously shown in Culex and Anopheles mosquitoes[20,21].

To investigate the biochemical and biological activities of these proteins, CxD7L1 and CxD7L2 mature cDNA sequences were codon-optimized for a eukaryotic cell expression system and engineered to contain a 6x-histidine tag in the C-terminal end followed by a stop codon. Both genes were subcloned into a VR2001-TOPO DNA cloning plasmid (Vical Inc.) as described by Chagas et al.[22]. Recombinant CxD7L1 and CxD7L2 proteins were expressed in human embryonic kidney (HEK293) cells and purified by affinity and size exclusion chromatography (Fig. 1b). The identities of purified recombinant proteins were confirmed by N-terminal and liquid chromatography tandem mass spectro-metry (LC/MS/MS sequencing). Both purified recombinant proteins migrated as single bands on Coomassie-stained precast polyacrylamide gels, and their apparent molecular weight (MW) in the gel corresponds to predicted MWs: 34.4 and 34.8 kDa for CxD7L1 and CxD7L2, respectively (Fig. 1c). Immunogenicity of both proteins in their recombinant forms was maintained, as they were recognized by the purified IgG antibodies from a rabbit immunized against C. quinquefasciatus salivary gland extract (SGE) (Supplementary Fig. 2a).

To perform immunolocalization experiments, specific anti-bodies against CxD7L1 and CxD7L2 were raised in rabbits. Because of the sequence similarity between these two proteins (34% identity), their antibodies showed cross-reactivity (Supple-mentary Fig. 2). To eliminate antibody cross-reactions and accurately identify D7 long form expression within salivary gland tissues, anti-CxD7L1 IgG was pre-adsorbed with CxD7L2 and anti-CxD7L2 IgG was pre-adsorbed with CxD7L1 (Supplemen-tary Fig. 2). As shown in Fig. 1d–f, CxD7L1 and CxD7L2 proteins are localized in the distal lateral and medial lobes of C. quinquefasciatus salivary glands, a pattern consistent with transcribed RNA of D7 long proteins in Ae. aegypti and An. gambiae[23,24].

A third D7 long protein was described by transcriptomic analysis of salivary glands of C. quinquefasciatus[7], as a 72.97% identical protein to CxD7L2 (BLASTp e-value: 3e$^{-167}$) with a unique insert of 20 amino acids (Supplementary Fig. 3a). We performed several experiments to evaluate whether this gene was transcribed and translated in the salivary glands. CxD7L3-specific sequence was amplified from gDNA extracted from male and female adult mosquitoes with similar levels but was absent in cDNA from female head and thorax (Supplementary Fig. 3b, c).

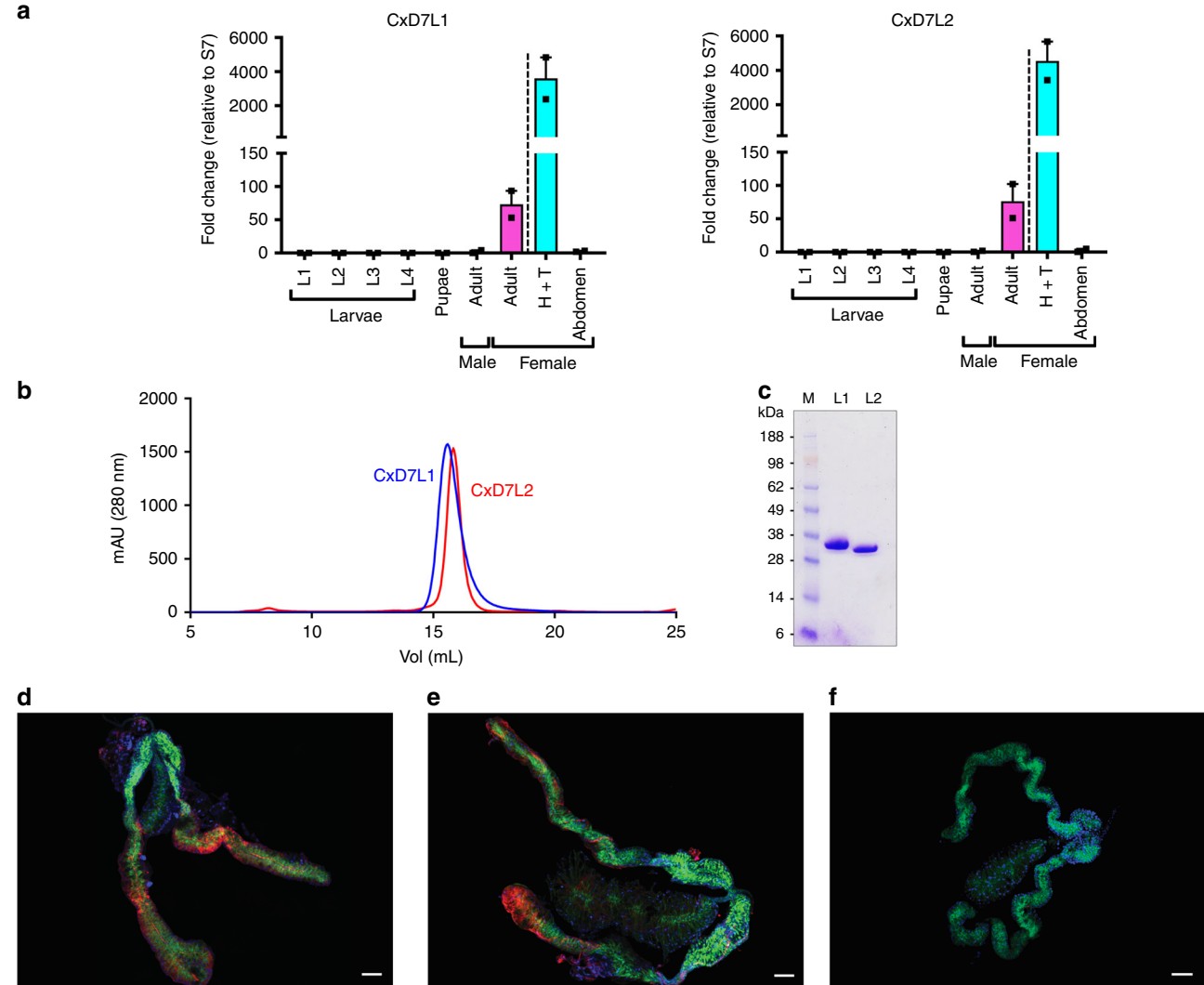

**Fig. 1 Characterization of *C. quinquefasciatus* salivary long D7 proteins. a** Gene expression analysis of *CxD7L1* and *CxD7L2* transcripts in different stages of *C. quinquefasciatus* mosquitoes. Relative abundance was expressed as the fold change using the 40S ribosomal protein S7 as the housekeeping gene. Larvae stage 1 (L1), larvae stage 2 (L2), larvae stage 3 (L3), larvae stage 4 (L4), pupae, male adult (reference sample), female adult, heads and thoraxes (H+T), and abdomens from female adult mosquitoes were analyzed separately. Two biological replicates and two technical duplicates were analyzed. Bars indicate the standard error of the means. **b** Purification of CxD7L1 (blue line) and CxD7L2 (red line) by size exclusion chromatography using Superdex 200 Increase 10/300 GL column. **c** Coomassie-stained NuPAGE Novex 4–12% Bis–Tris gel electrophoresis (*n* = 1) of recombinant proteins CxD7L1 and CxD7L2 (1.5 μg). SeeBlue Plus2 Pre-stained was used as the protein standard (M). **d**, **e** Immunolocalization of CxD7L1 and CxD7L2 proteins in the salivary glands of *C. quinquefasciatus*. Salivary glands were incubated with rabbit IgG anti-CxD7L1 (**d**), anti-CxD7L2 (**e**), and further stained with anti-rabbit IgG Alexa Fluor 594 antibody shown in red. Proteins of interest were localized in the medial and distal regions of the lateral lobes of *C. quinquefasciatus* salivary glands. As a control, salivary glands were incubated with anti-rabbit IgG AF594 alone (**f**). Nucleic acids were stained by DAPI (blue) and the actin structure of salivary glands was stained using Phalloidin Alexa 488 (green). Four independent experiments were performed with 1–2 glands imaged per experimental group. Scale bar = 50 μm. Source data are provided as a Source Data file.

To evaluate if CxD7L3 protein was present in the salivary glands of *C. quinquefasciatus*, mass spectrometry of SGE was performed. These experiments showed that the unique 20 amino acid insertion that characterized CxD7L3 was not detected in the SGE (Supplementary Fig. 3d).

Our quantification analysis showed that the amount of CxD7L1 and CxD7L2 present in the salivary glands of 5-day-old *C. quinquefasciatus* was 31.33 ± 6.15 and 35.97 ± 9.59 ng, respectively. To investigate the amount of D7 proteins released during a mosquito bite, the same batch of mosquitoes were allowed to feed on chicken and salivary glands were dissected immediately after blood-feeding. As a result of probing, CxD7L1 and CxD7L2 proteins were 22.13 and 26.83%, respectively,

reduced in the salivary glands after blood-feeding. We deducted that the injected amounts of protein during feeding were 6.93 ± 1.34 and 9.65 ± 5.59 ng for CxD7L1 and CxD7L2, respectively. These estimated values are similar to the ones obtained by quantifying the proteins of oil-induced saliva (6.45 ± 1.7 ng of CxD7L1 and 4.28 ± 0.98 ng of CxD7L2, per salivating mosquito).

**CxD7L1 binds adenine-nucleosides and nucleotides.** Previous work demonstrated that members of the D7-related protein family can bind to biogenic amines and eicosanoids[10,11,13,14]. The binding abilities of CxD7L1 were tested with a wide panel of pro-hemostatic compounds including biogenic amines, nucleic acids,

| Table 1 Thermodynamic parameters of CxD7L1 protein by ITC. | | | | | |
|---|---|---|---|---|---|
| Ligand | Binding | Stoichiometry | $\Delta H$, cal/mol ± SE | $T\Delta S$, cal/mol/deg | $K_D$, nM |
| 5′-ATP | Yes | 0.91 | −1.72E4 ± 277.3 | −22.20 | 30.77 |
| 5′-ADP | Yes | 0.90 | −1.80E4 ± 416.9 | −25.00 | 32.68 |
| 5′-AMP | Yes | 0.92 | −1.93E4 ± 560.8 | −31.20 | 77.52 |
| Adenosine | Yes | 0.85 | −1.15E4 ± 668.0 | −31.20 | 312.50 |
| Adenine | Yes | 1.00 | −9.60E3 ± 1.97E3 | −5.35 | 1760.56 |
| 5′-GTP | No | N/A | N/A | N/A | N/A |
| 5′-TTP | No | N/A | N/A | N/A | N/A |
| 3′-AMP | No | N/A | N/A | N/A | N/A |
| cAMP | No | N/A | N/A | N/A | N/A |
| PolyP | No | N/A | N/A | N/A | N/A |

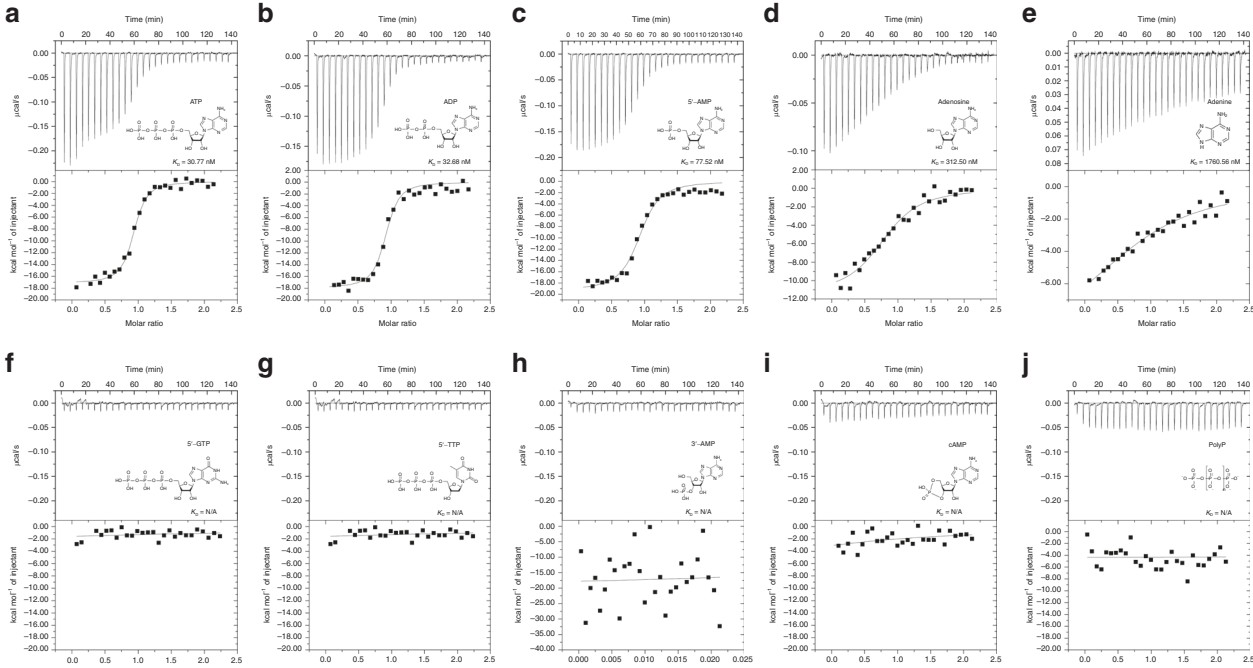

**Fig. 2 Binding of nucleosides and related molecules to CxD7L1 by ITC.** Binding experiments were performed on a VP-ITC microcalorimeter. Assays were performed at 30 °C. The upper curve in each panel shows the measured heat for each injection, while the lower graph shows the enthalpies for each injection and the fit to a single-site binding model for calculation of thermodynamic parameters. Titration curves are representative of two measurements. Panels (**a–e**) show adenine nucleosides or nucleotides that bind CxD7L1: adenosine 5-triphosphate (**a**), adenosine 5-diphosphate (**b**), adenosine 5-monophosphate (**c**), adenosine (**d**), and adenine (**e**). In panels (**f–j**), other purine and pyrimidine nucleotides and related substances showed no binding to CxD7L1: guanosine 5-triphosphate (**f**), thymidine 5-triphosphate (**g**), adenosine 3-monophosphate (**h**), cyclic adenosine monophosphate (**i**), and polyphosphate (**j**). The insets show the names and chemical formulas for these compounds.

and proinflammatory lipids using isothermal titration calorimetry (ITC). In contrast to its D7 orthologs in *Aedes* and *Anopheles* mosquitoes, CxD7L1 does not bind biogenic amines such as serotonin, nor the pro-inflammatory lipids LTB₄ and LTD₄ or the stable analog of TXA₂, U-46619 (Supplementary Fig. 4). However, CxD7L1 has evolved to bind adenine-nucleosides and nucleotides with high affinity (Table 1 and Fig. 2), a novel function in a D7-related protein.

Our biochemical characterization shows that CxD7L1 specifically binds the purine nitrogenous base adenine, its nucleoside (adenosine), and nucleotide derivates: AMP, ADP, and ATP, with the highest affinity to ATP and ADP (Fig. 2a–e). The binding is adenine-specific, as no binding was observed with other purine or pyrimidine nucleotides such as GTP or TTP (Fig. 2f, g). Although adenine is essential for binding, CxD7L1 did not bind to adenosine 3′-monophosphate (3′-AMP) or cyclic AMP (Fig. 2h, i), highlighting the importance of the phosphate group position in

binding stabilization. Interaction between CxD7L1 protein and phosphate alone was ruled out as polyphosphate (sodium phosphate glass type 45) did not bind to the protein in ITC experiments (Fig. 2j). Furthermore, CxD7L1 did not bind to inosine (Supplementary Fig. 4), an intermediate metabolite in the purine metabolic pathway.

**CxD7L2 binds to biogenic amines and eicosanoids.** A detailed analysis of binding activities using ITC shows that CxD7L2 has comparable ligand-binding capabilities as previously described in *Aedes* long and *Anopheles* long and short D7 proteins (Table 2 and Fig. 3)[10,11,13]. CxD7L2 tightly binds serotonin ($K_D = 7.5$ nM) and other biogenic amines, including histamine and epinephrine, with lower affinities. It does not, however, bind norepinephrine. CxD7L2 also binds the cysteinyl leukotrienes, LTC₄, LTD₄, and LTE₄ with a stoichiometry of 1:1 all with similar binding affinities

**Table 2 Thermodynamic parameters of CxD7L2 protein by ITC.**

| Ligand | Binding | Stoichiometry | $\Delta H$, cal/mol $\pm$ SE | $T\Delta S$, cal/mol/deg | $K_D$, nM |
|---|---|---|---|---|---|
| Serotonin | Yes | 1.37 | $-1.63E4 \pm 171.2$ | $-16.50$ | 7.46 |
| Histamine | Yes | 0.97 | $-1.31E4 \pm 579.4$ | $-14.00$ | 383.14 |
| Epinephrine | Yes | 0.94 | $-5.79E4 \pm 513.8$ | 11.30 | 226.24 |
| LTB$_4$ | No | N/A | N/A | N/A | N/A |
| LTC$_4$ | Yes | 1.07 | $-2.24E4 \pm 621.2$ | $-42.80$ | 151.75 |
| LTD$_4$ | Yes | 0.98 | $-1.53E4 \pm 812.7$ | $-19.40$ | 156.49 |
| LTE$_4$ | Yes | 1.07 | $-1.62E4 \pm 561.8$ | $-22.40$ | 158.73 |
| Arachidonic acid | Yes | 1.29 | $-6.66E3 \pm 578.4$ | $-5.33$ | 1083.42 |
| U-46619 | Yes | 0.99 | $-6.06E3 \pm 474.4$ | 7.58 | 934.58 |
| 5′-ATP | No | N/A | N/A | N/A | N/A |
| 5′-ADP | No | N/A | N/A | N/A | N/A |

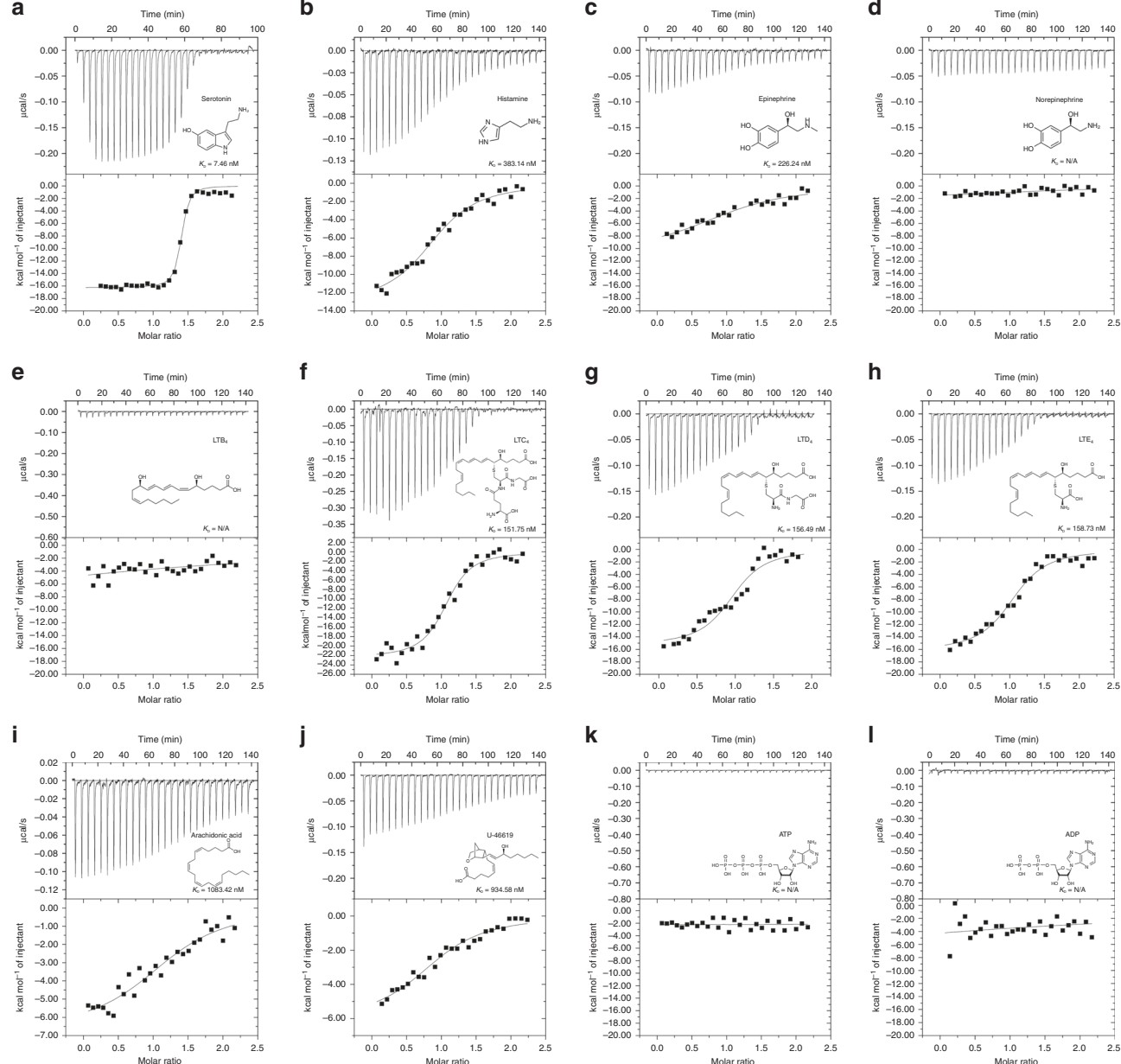

**Fig. 3 Binding of biogenic amines and eicosanoids to CxD7L2 by ITC.** Binding experiments were performed on a VP-ITC microcalorimeter. The upper curve in each panel shows the measured heat for each injection, while the lower graph shows the enthalpies for each injection and the fit to a single-site binding model for calculation of thermodynamic parameters. Titration curves are representative of two measurements. Panels: serotonin (**a**), histamine (**b**), epinephrine (**c**), norepinephrine (**d**), LTE$_4$ (**e**), LTC$_4$ (**f**), LTD$_4$ (**g**), LTE$_4$ (**h**), arachidonic acid (**i**), TXA$_2$ analog U-46619 (**j**), ATP (**k**), and ADP (**l**). The insets show the names and chemical formulas for these compounds.

($K_D$ = 151.8, 156.5, and 158.7 nM, respectively, Table 2 and Fig. 3). CxD7L2 also binds arachidonic acid and U-46619 with lower affinities ($K_D$ = 1083.42 and 934.6 nM, respectively) when compared to the cysteinyl LT. No binding to LTB$_4$, ATP, or ADP was detected (Fig. 3).

To gain insights into the mechanism of CxD7L2 binding to biogenic amines and eicosanoids, the N-terminal and C-terminal domains were independently cloned and expressed in *Escherichia coli*. Only the C-terminal domain of CxD7L2 (CxD7L2-CT) was successfully purified and analyzed in parallel with the full-length protein by ITC. Similar to the full-length CxD7L2 protein, CxD7L2-CT binds to serotonin with high affinity ($K_D$ = 1.5 nM, $N$ = 1.06, $\Delta H$ = 4.31E4 ± 460 cal/mol; for CxD7L2-serotonin, see Table 2). We concluded that CxD7L2-CT is responsible for the serotonin binding capacity displayed by the full-length protein. Since we were unable to produce the CxD7L2 N-terminal domain as a non-aggregated protein, a saturation study was designed to indirectly investigate the binding specificity of this domain. For this experiment, CxD7L2 protein was saturated with serotonin and titrated with LTD$_4$. The calculated binding parameters for CxD7L2 titrated with LTD$_4$ in the absence or presence of serotonin remained similar ($K_D$ = 156.8 nM, $N$ = 0.93, $\Delta H$ = −2.21E4 ± 924.6 cal/mol; for CxD7L2-LTD$_4$, see Table 2). These results demonstrate that lipids and biogenic amines bind to the CxD7L2 protein independently through different binding pockets, with lipids binding to the N-terminal pocket and biogenic amines to the C-terminal pocket, similar to the binding mechanism of AeD7 protein from *Ae. aegypti*[11].

**Crystal structure of *Culex quinquefasciatus* CxD7L1.** To further characterize the mechanism of the adenine nucleoside/nucleotide D7 binding, we solved the crystal structure of CxD7L1 in complex with ADP. A crystal of CxD7L1 that belonged to $I2_12_12_1$ space group and diffracted to 1.97 Å resolution was used to collect a data set (Table 3). The coordinates and structure factors have been deposited in the Protein Data Bank under the accession number 6V4C.

The CxD7L1 protein fold consists of 17 helical segments stabilized by five disulfide bonds linking C18 with C51, C47 with C104, C154 with C186, C167 with C295, and C228 with C242 (Fig. 4a, b). The structure revealed that the ligand-binding site is located between the N-terminal and C-terminal domains (Fig. 4a–e). All hydrogen bond donors and acceptors present in the adenine ring (N1, N3, and N7 are acceptors, and N6 is a donor) are interacting with the protein resulting in stable binding. The residues involved in binding ADP or stabilizing the binding pocket are S130, R133, Y137, K144, K146, N265, Y266, S263, S267, and R271 (Fig. 4e). Residues Y137, K144, and Y266 bind to the adenine ring. The hydroxyl group of Y137 forms a bidentate hydrogen bond with the N6 and N7 of the adenine ring. The carbonyl oxygen of K144 forms a hydrogen bond with the amino nitrogen N6 of the adenine ring, while the NZ of K144 is involved in two hydrogen bonds, one with N1 from the adenine ring, and the other with the carbonyl oxygen of S263. The hydrogen bond with the carbonyl oxygen of S263 fixes NZ of the K144 in a position that allows it to bind the adenine ring. The amide nitrogen of Y266 binds N3 of the adenine ring and its side chain stacks partially on top of the base of ADP which provides a favorable van der Waals contribution to the CxD7L1-ADP interaction. As we go further along the ADP molecule, S267 interacts strongly with and fixes the ribose ring of ADP with its hydroxyl group involved in two hydrogen bonds with both O2′ and O3′. In addition, the ribose oxygen O2′ forms a hydrogen bond with a water molecule and ND2 of N265 binds to O5′ of the sugar. The side chain of S130 forms a hydrogen bond with the

oxygen of the beta phosphate. R271 makes a hydrogen bond to N265 so that it is positioned favorably to engage in electrostatic interaction with the alpha phosphate. K146 is also in a location that can potentially be involved in electrostatic interaction with the alpha phosphate. R133 forms two salt bridges with the beta phosphate of ADP, with NH1 and NH2 of R133 binding to O1B and O3B of ADP respectively, which may explain the similar binding affinities between ATP and ADP and the lower affinity of AMP, which lacks the beta phosphate.

Although the superposition of structures of CxD7L1, AeD7 (PDB: 3DZT), and AnStD7L1 (PDB: 3NHT) showed a similar overall structure (Fig. 5a), the protein sequences only share 20% amino acid identity and some of the essential residues involved in the lipid and biogenic amine binding are missing in CxD7L1 (Supplementary Fig. 1). Moreover, CxD7L1 showed a completely different electrostatic surface potential when compared to *Ae. aegypti* D7L and *An. stephensi* AnStD7L1, which may contribute to the differences in their binding capacity. The amino acids that constitute the ADP binding pocket in CxD7L1 create a strongly negative surface, showing an inverted pattern of amino acid charges that completely change the nature of the binding pockets (Fig. 5b). Some, but not all of the residues involved in ADP binding were conserved in other D7 homologs (Supplementary Fig. 1). Although most of the residues were present in D7 long proteins from *Culex tarsalis* (Supplementary Fig. 5) no experimental data is available showing that D7L1 from this mosquito retains the ADP binding capacity.

**CxD7L1 and *Culex quinquefasciatus* bites reduce ADP levels.** Because our ITC experiments indicate that CxD7L1 binds ADP with high affinity, we investigated whether this salivary protein

| Table 3 Data collection and refinement statistics. | |
|---|---|
| | **CxD7L1-ADP complex** |
| **Data collection** | |
| Space group | $I2_12_12_1$ |
| Cell dimensions | |
| $a, b, c$ (Å) | 76.66, 84.32, 132.07 |
| Resolution (Å) | 71.07–1.97 (2.02–1.97)[a] |
| $R_{merge}$ (%)[b] | 6.6 (64.3) |
| $I/\sigma I$ | 12.19 (2.35) |
| Completeness (%) | 99.1 (100) |
| Redundancy | 5.91 (5.95) |
| **Refinement** | |
| Resolution (Å) | 39.02–1.97 (2.02–1.97) |
| No. reflections | 29,350 |
| $R_{work}/R_{free}$ (%) | 21.36/23.49 (26.22/30.42) |
| No. atoms | |
| Protein | 2255 |
| Ligand (additives) | 77 |
| Water | 121 |
| Metal (Zn$^{2+}$) | 2 |
| $B$-factors (Å$^2$) | |
| Protein | 57.51 |
| Ligand (additives) | 60.27 |
| Water | 48.28 |
| Metal (Zn$^{2+}$) | 50.96 |
| RMS deviations | |
| Bond lengths (Å) | 0.006 |
| Bond angles (°) | 0.800 |

[a]Values in parentheses are for the highest-resolution shell.
[b]$R$-merge($I$) = $\sum_{hkl}(\sum_i|I_i(hkl)) - \langle I(hkl)\rangle|/\sum_{hkl}\sum_i I_i(hkl)$, where $I_i(hkl)$ is the intensity of the *i*th observation of a reflection with indices ($hkl$), including those of its symmetry mates, and $\langle I(hkl)\rangle$ is the corresponding average intensity for all *i* measurements.

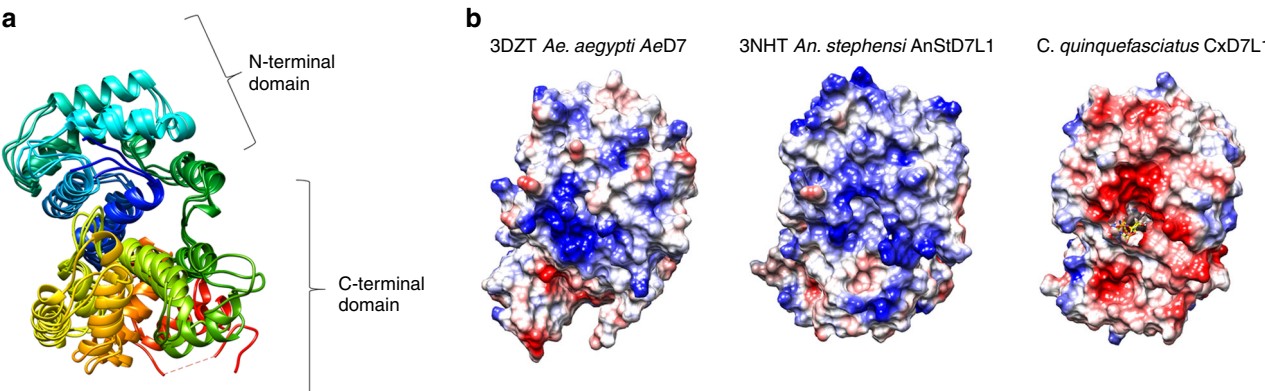

**Fig. 4 Structure of CxD7L1 in complex with ADP. a** Ribbon representation of CxD7L1-ADP structure. The 17 α-helices are labeled A–Q. **b** Several views of CxD7L1 differing by rotations of 90° around the *y*-axis. N-terminal and C-terminal are colored in blue and green, respectively. ADP is shown as a stick model in magenta and disulfide bonds in orange. **c** Electron density map covering ADP. CxD7L1 protein is colored in green. **d** Inset from (**c**) is shown. Amino acid residues of CxD7L1 involved in ADP binding are colored in green (**e**). Stereo view of the binding pocket of the CxD7L1-ADP complex showing the $2F_o - F_c$ electron density contoured at 1σ covering the ligand. All residues within a 3.6 Å distance from the ADP are shown. Hydrogen bonds are colored in yellow. GOL-402 is a glycerol molecule that is in the area of the phosphate group. R271 was not included in the figure as it is not within 3.6 Å and it binds N265, but it does not bind ADP directly.

**Fig. 5 Multiple sequence superposition and electrostatic potential of *Culex* D7 proteins. a** Superposition of CxD7L1, *Ae. aegypti* AeD7 (PDB ID: 3DZT) and *An. stephensi* AnStD7L1 (PDB ID: 3NHT) shows a similar overall helix structure. Rainbow coloring pattern shows the N-terminal in blue and the C-terminal in red. **b** Electrostatic potential of 3DZT, 3NHT, and CxD7L1 generated by Coulombic Surface Coloring (Chimera software) with blue being positive and red being negative. ADP is represented as a stick model.

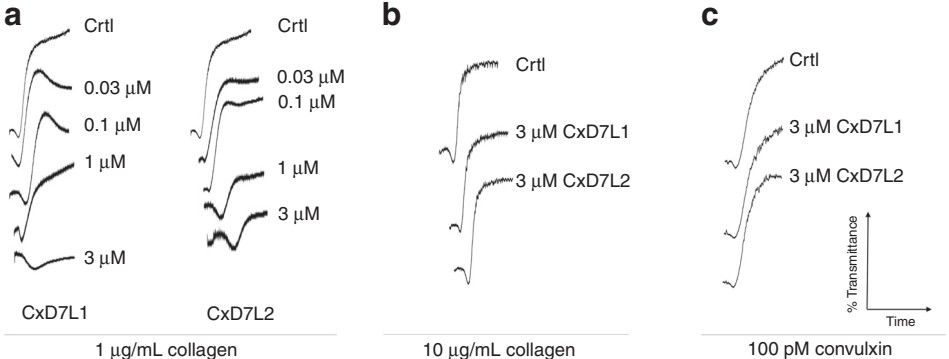

**Fig. 6 Effect of CxD7 proteins on platelet aggregation induced by collagen or convulxin.** Prior to the addition of the agonist, platelet-rich human plasma was incubated for 1 min with either PBS (Crtl) or with the recombinant proteins at the concentrations shown. Aggregometer traces were measured at 37 °C from stirred platelets suspensions on a Chrono-Log platelet aggregometer model 700 for 6 min. An increase of light transmittance over time indicates platelet aggregation. **a** CxD7L1 and CxD7L2 concentration-dependent inhibition of platelet aggregation induced by low doses of collagen (1 μg/mL). CxD7L1 and CxD7L2 failed to inhibit platelet aggregation induced by **b** high doses of collagen (10 μg/mL) and **c** GPVI agonist convulxin (100 pM). Graphs are representative of two measurements. Source data are provided as a Source Data file.

could scavenge ADP at a mosquito bite. We exposed mice ears to *C. quinquefasciatus* bites and immediately after biting, ears were removed for ADP determination. Non-bitten ears contained $218.7 \pm 15.36$ pmol/μL of ADP while bitten ears showed a reduction of 39.45% in ADP concentration ($132.4 \pm 18.79$ pmol/μL), as shown in Supplementary Fig. 6. As a confirmation of the scavenging properties of CxD7L1 towards ADP, we also performed the ADP quantification in samples incubating a known amount of ADP and equimolar concentrations of CxD7L1. We observed a drastic reduction of 99.99% in the ADP levels when the protein was present (1.0354686 pmol/μL in the ADP only sample compared to 0.0001457 pmol/μL in ADP and CxD7L1 combined).

**CxD7L1 and CxD7L2 play a role in platelet aggregation.** Because CxD7 long forms bind platelet aggregation agonists, we examined their ability to interfere with platelet aggregation in ex vivo experiments. At low concentrations of collagen (1 μg/mL), we saw the classical collagen induction trace, where there is a delay of the platelet shape change due to the release of secondary mediators and observed as the initial decrease of light transmittance. There was a clear dose-dependent inhibition of platelet aggregation by both CxD7L1 and CxD7L2 (Fig. 6a). Neither CxD7L1 nor CxD7L2 interfered with platelet aggregation induced by high doses of either collagen (Fig. 6b) or convulxin (Fig. 6c), an agonist of the platelet GPVI collagen receptor which induces platelet aggregation independently of secondary mediators.

We also investigated the anti-platelet aggregation activity of CxD7L1 and CxD7L2 using ADP as an agonist. ADP plays a role in the initiation and extension of the aggregation cascade. When ADP was added at concentrations below the threshold for platelet aggregation (0.5 μM), only platelet shape change was observed (control trace, Fig. 7a). Preincubation of platelets with CxD7L1 prevented this shape change. With higher doses of ADP (1 μM), platelet aggregation was inhibited in the presence of 3 μM CxD7L1 (Fig. 7a). At high doses of ADP (10 μM), 3 μM of CxD7L1 was insufficient to inhibit platelet aggregation, confirming the nature of the inhibition by scavenging the mediator. The addition of CxD7L2 did not show any effect in aggregation initiated via ADP at any dose, confirming that CxD7L2 does not target ADP (Fig. 7a).

We also used U-46619, the stable analog of TXA$_2$ and widely accepted for platelet aggregation studies[13,14,26,27]. When platelets are activated, TXA$_2$ is synthesized from arachidonic acid released

from platelet membrane phospholipids. TXA$_2$ is an unstable compound and cannot be evaluated directly as a platelet aggregation agonist ex vivo. CxD7L2 inhibited U-46619-induced platelet aggregation in a dose-dependent manner. However, platelet shape change requires minimal concentrations of TXA$_2$, and it was not prevented by CxD7L2 (Fig. 7b). Shape change was only abolished in the presence of 1 μM SQ29,548, a specific antagonist of the TXA$_2$ receptor (Fig. 7b). This result is supported by our biochemical data showing that CxD7L2 binds directly to U-46619 in vitro (Fig. 3j). However, we do not know whether this binding is retained in vivo.

To verify that this protein binds the biological active TXA$_2$ ex vivo, we induced platelet aggregation with its biosynthetic precursor, arachidonic acid, so that TXA$_2$ would be released by platelets. CxD7L2 inhibited platelet aggregation induced by arachidonic acid only at high doses of protein (6 μM, Fig. 7b), most likely due to the low binding affinity observed for U-46619 and arachidonic acid (Table 2). To further investigate whether this effect was a result of a direct sequestering of TXA$_2$ by CxD7L2, we pre-incubated platelets with indomethacin, a cyclooxygenase-1 inhibitor, that prevents TXA$_2$ biosynthesis. We observed almost no inhibition of low dose collagen-induced platelet aggregation in the presence of CxD7L2 (Fig. 7b), indicating that the anti-platelet aggregation activity of CxD7L2 is mediated by TXA$_2$ binding.

CxD7L1 inhibits platelet aggregation induced by U-46619 in a dose-dependent manner (Fig. 7b). CxD7L1 does not bind U-46619 (Supplementary Fig. 4), but it tightly binds ADP (Fig. 2b and Table 1). Platelet aggregation triggered by U-46619, arachidonic acid, and low doses of collagen is highly dependent on ADP[28]. As a confirmation of this dependence, CxD7L1 inhibits platelet aggregation stimulated by either U-46619 or arachidonic acid as effectively as the antagonist of the TXA$_2$ receptor SQ29,548. CxD7L1 also prevented aggregation initiated by low dose of collagen in indomethacin-treated platelets (Fig. 7b).

Serotonin acts as a potentiator of platelet agonists such as ADP or collagen. Alone, serotonin can initiate platelet aggregation, but in the absence of a more potent agonist, the platelets eventually disaggregate (Supplementary Fig. 7a). CxD7L2 tightly binds serotonin (Fig. 3a). Therefore, the initiation of aggregation produced by serotonin was completely abolished in the presence of equimolar concentrations of the recombinant protein (Supplementary Fig. 7a). However, when a higher dose of serotonin was used (10 μM), CxD7L2 was unable to sequester all the serotonin,

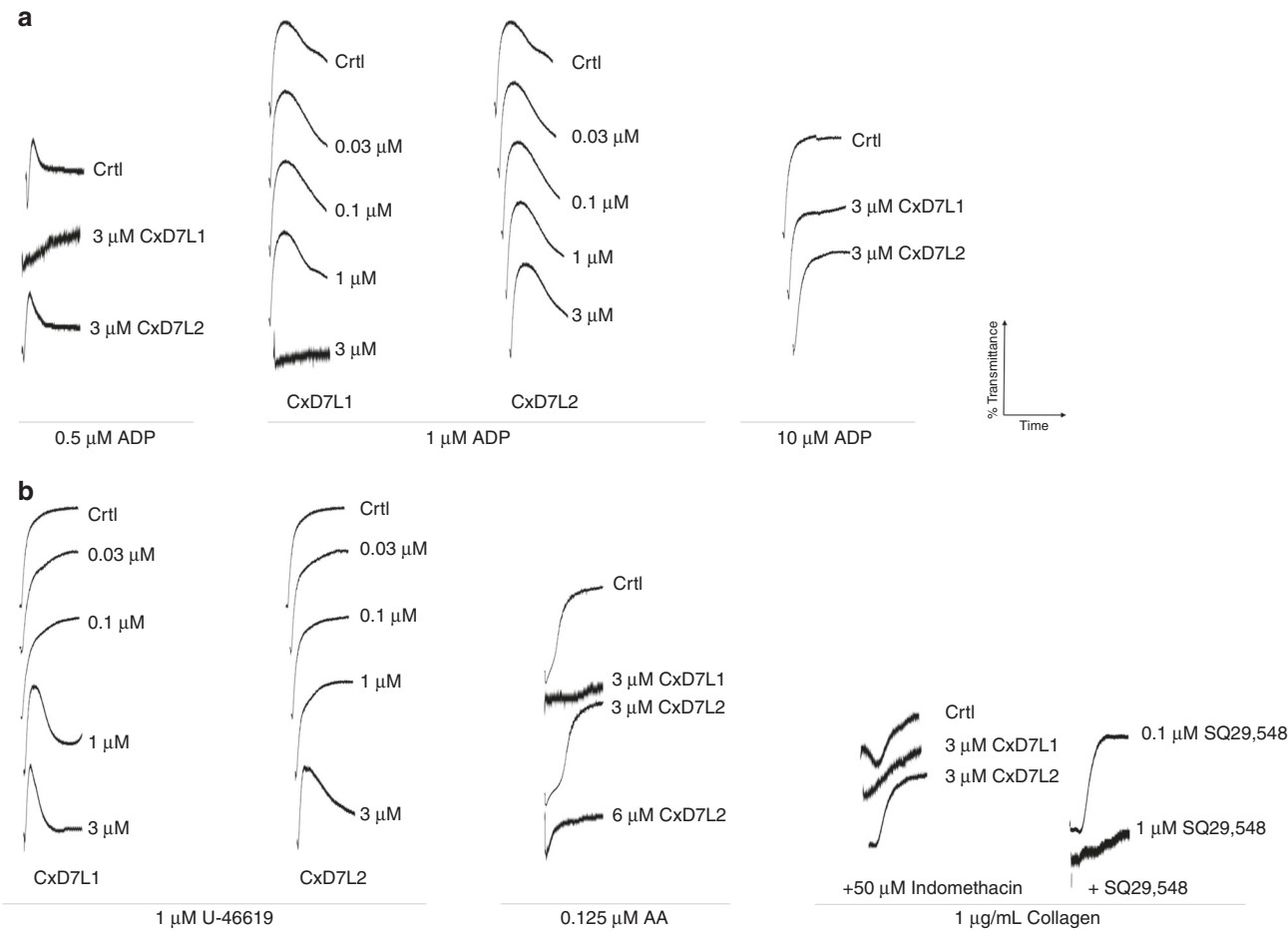

**Fig. 7 Effect of CxD7 proteins on platelet aggregation induced by secondary mediators.** Prior to the addition of the agonist, platelet-rich human plasma was incubated for 1 min either with PBS (Crtl) or with the recombinant proteins, or SQ29,548 at the concentrations shown. Aggregometer traces were measured at 37 °C from stirred platelets suspensions on a Chrono-Log platelet aggregometer model 700 for 6 min. An increase of light transmittance over time indicates platelet aggregation. **a** Platelet aggregation traces using different concentrations of ADP (0.5, 1, and 10 μM) as aggregation agonist. **b** Platelet aggregation traces using 1 μM U-46619, 0.125 μM arachidonic acid (AA), or low collagen concentration (1 μg/mL). Graphs are representative of two measurements. Source data are provided as a Source Data file.

resulting in no observed inhibition of platelet aggregation (Supplementary Fig. 7a). When serotonin and low doses of collagen were used as aggregation agonists, CxD7L1 partially prevented aggregation, presumably due to its ADP binding, while CxD7L2-serotonin binding resulted in full inhibition of platelet aggregation (Supplementary Fig. 7b). Serotonin also potentiated aggregation initiated by low doses of ADP (Supplementary Fig. 7c). When platelets were incubated with CxD7L2, the synergistic effect of serotonin and ADP was abolished (Supplementary Fig. 7c). CxD7L1, as a potent ADP-binder, completely abrogated platelet aggregation initiated by serotonin and ADP combined. In addition, CxD7L2 partially prevented aggregation initiated by serotonin and epinephrine (Supplementary Fig. 7d).

**CxD7L1 and CxD7L2 extended tail vein bleeding time.** To evaluate the overall role of proteins on the host hemostasis, tail vein bleeding time experiments were performed in mice. After laceration, the tail was immersed in tubes containing saline or CxD7 proteins at different concentrations. Both CxD7 proteins showed a dose-dependent extension of mouse bleeding time as a result of impairment of hemostasis. Although the trend for both proteins is similar, only CxD7L1 shows a statistically significant

difference when compared to the mouse tails immersed in only saline (Supplementary Fig. 8).

## Discussion

During a bite, arthropod salivary proteins are injected into the host skin to counteract host hemostatic mediators. In this work, we characterized the structure and function of the salivary D7 long proteins from *C. quinquefasciatus* mosquitoes and described a mechanism of platelet aggregation inhibition for a D7 salivary protein.

CxD7L1 and CxD7L2 were found to be localized in the distal-lateral and medial lobes of *C. quinquefasciatus* salivary glands. Salivary proteins have been shown to accumulate in the salivary glands forming distinct spatial patterns[23]. Although the relevance of distinct protein localization is not yet well understood, it supports the hypothesis of functionally-distinct regions within mosquito salivary glands. Salivary proteins related to sugar-feeding, nectar-related digestion, and bactericidal functions are localized in the proximal-lateral lobes, while proteins involved in blood-feeding, such as CxD7L1 and CxD7L2, are localized in the medial or distal-lateral lobes. More research is required to understand the implications of the salivary protein compartmentalization and viral infection of the glands. Besides CxD7L1

and CxD7L2, another D7 long protein (CQ_LD7_3 = CxD7L3) was described in a transcriptome of salivary glands from *C. quinquefasciatus*[7]. Its predicted amino acid sequence shares a 72.97% identity with CxD7L2 and presents a unique insert of 20 amino acids flanked by glycine and proline residues, often associated with beginning and end of protein loops. PCR experiments showed that the specific region for CxD7L3 is present in the gDNA of adult mosquitoes, but it is not transcribed to mRNA in the salivary glands. Additionally, the unique 20 amino acid insertion that characterized CxD7L3 was not detected in SGEs by mass spectrometry. Taken all these experiments together, we believe there is a high probability that CxD7L3 is a result of an error in the transcriptomic annotation of *C. quinquefasciatus* salivary glands. This agrees with a recent bioinformatic analysis of six *Anopheles* species genomes that states that 46% of the salivary transcripts were wrongly predicted[29].

D7 proteins are widely distributed in the saliva of hematophagous Nematocera, including mosquitoes, black flies, biting midges, and sand flies[8]. D7 salivary proteins antagonize the hemostasis mediators through a non-enzymatic, non-receptor-based mechanism by binding and sequestering several host hemostasis mediators[8,10,11,13,14]. Biogenic amines play important physiological roles in host hemostasis. Serotonin is released from platelet granules upon activation and acts as a weak platelet aggregation agonist. Serotonin and histamine increase vascular permeability and induce host sensations of pain and itch[30]. The catecholamines norepinephrine and epinephrine stimulate vasoconstriction by directly acting on adrenoreceptors[12]. Binding of biogenic amines by mosquito D7 proteins has been previously reported in the literature, highlighting the importance of removing these mediators at the bite site[10,11,31]. Binding affinities for the different amines vary, as D7 proteins have become highly specialized for specific ligands[10,11,13,14]. CxD7L2 tightly binds serotonin and epinephrine in the same range as the short D7 proteins from *An. gambiae* and AeD7 from *Ae. aegypti*[10,11]. However, it showed lower affinity for histamine and did not bind norepinephrine. Like AeD7 from *Ae. aegypti*[11], CxD7L2 is multifunctional and was able to bind biolipids through its N-terminal domain and biogenic amines through its C-terminal domain, as confirmed by ITC experiments. CxD7L2 binds cysteinyl leukotrienes ($LTC_4$, $LTD_4$, and $LTE_4$) with similar affinities. Cysteinyl leukotrienes are potent blood vessel constrictors and increase vascular permeability[32]. The cysteinyl residue appears to play a role in lipid binding, as calorimetry experiments with lipids lacking a cysteinyl residue such as $LTB_4$ showed no binding. Residues involved in bioactive lipid binding were conserved between CxD7L2 and the D7 proteins from *An. stephensi* and *Ae. aegypti* (AnStD7L1 and AeD7). Interestingly, a tyrosine residue at position 52 is present in *Culex* D7 long proteins and has been correlated to the ability to stabilize the binding of the $TXA_2$ mimetic (U-46619) in *An. stephensi*[13]. This residue is absent in the *Ae. aegypti* D7 protein that does not bind U-46619[11]. This might explain the ability of CxD7L2 to bind cysteinyl leukotrienes and U-46619. Additionally, several residues known to be involved in the biogenic amine-binding were conserved in *Culex* D7 long proteins, for which the biogenic amine binding capability of CxD7L2 may be accounted.

Although CxD7L1 retains some amino acids involved in biogenic amine or lipid binding, ITC data showed that this protein lacks binding capacities typical of D7 proteins. Rather, CxD7L1 binds adenine nucleosides and nucleotides. Our crystallographic data clearly confirms our binding results. The nature of the binding pocket demonstrates specificity for the adenine ring. The hydrogen bonds between the adenine ring and residues Y137, K144, and Y266 determine the specificity for adenine and the lack of binding to other nucleotides with other nitrogenous bases

(5′-GTP and 5′-TTP). Similarly, S267 and N265 of CxD7L1 are involved in binding to the ribose, which is possible when the phosphate group occupies position 5′ but not position 3′ or the cyclic form, as shown by calorimetry experiments. R133 binds to the oxygen of the beta phosphate of ADP which may explain the similar binding affinities for both ATP and ADP while affinity for AMP is lower as it lacks the beta phosphate.

The scavenging mechanism of action requires a high concentration of salivary protein at the bite site. As D7 proteins bind their ligands in a 1:1 stoichiometric ratio, they must be in equimolar concentrations with the mediators, which range from 1 to 10 μM for histamine, serotonin, or ADP[8]. This may explain why D7 salivary proteins are one of the most abundant components of the salivary glands. We determined that the amount of CxD7L1 and CxD7L2 present in the SGE was 31.33 and 35.97 ng, respectively. Our quantification data is within range with previous studies from *Ae. aegypti* in which authors quantified other majoritarian salivary proteins such as Aegyptin[33] or apyrase[34], obtaining 36.35 and 24 ng per salivary gland pair, respectively per salivary gland. Besides, we determined the amount of D7 proteins in the secreted saliva by two different methods. Both approaches gave similar results, confirming that the protein amount deducted from the SGE before and after blood-feeding corresponded well to what is present in the secreted saliva. After blood-feeding, we observed 22.13 and 26.83% reduction, respectively, of CxD7L1 and CxD7L2 protein amounts in the SGE. In *Ae. aegypti*, other authors have found a greater reduction of either total salivary protein or apyrase activity after blood-feeding (58 and 46% reduction, respectively)[35]. On the other hand, our approach to quantify ADP in mouse skin before and after mosquito bites showed that the bites reduced 37.45% of the amount of ADP. In our ADP determinations, the effect of salivary apyrase, also present in the saliva of *C. quinquefasciatus*, was minimized by performing all assays in the presence of ethylenediaminetetraacetic acid (EDTA) which chelates $Ca^{+2}$ and other metals necessary by the apyrases to cleave ADP[36]. Therefore, we can attribute the 37.45% ADP reduction to CxD7L1.

Platelet aggregation occurs within seconds of tissue injury, restricting blood flow and creating a platelet plug that reduces blood feeding success. Exposure of circulating platelets to collagen from the subendothelial matrix or thrombin leads to the formation of a platelet monolayer that supports subsequent adhesion of activated platelets to each other[12,37]. At low concentrations of collagen, ADP and $TXA_2$ play an important role on the extension and amplification step of the platelet plug formation. Upon platelet activation, mediators secreted by platelets bind to G protein-coupled receptors in platelet membranes, rapidly amplifying the aggregation signal in a positive feedback response[38]. However, at high concentrations, collagen acts as a strong agonist of the GPVI receptor on platelet surface, which induces platelet aggregation in an independent manner of ADP or $TXA_2$ secretion[37]. Both CxD7L1 and CxD7L2 proteins showed a potent inhibitory effect on platelet aggregation, explained by distinct mechanisms. CxD7L2 inhibits platelet aggregation in the classical mechanism observed in other eicosanoid-scavenging salivary proteins[13,14,26,39,40]. CxD7L2 inhibits low dose collagen-induced platelet aggregation in a dose-dependent manner but did not affect aggregation induced by high doses of collagen or convulxin. These findings indicate that CxD7L2's inhibitory effect on platelet aggregation is dependent on secondary mediators and does not interfere with collagen directly. CxD7L2 showed a low binding affinity for U-46619, the stable analog of $TXA_2$ (934.58 nM), and its precursor, arachidonic acid (1083.42 nM) which might explain the high doses needed to neutralize the aggregation induced by arachidonic acid. CxD7L2 also binds serotonin and epinephrine which act as weak platelet agonists alone, but are important as

they reduce the threshold concentrations of other agonists for platelet aggregation, as previously observed for the biogenic amine-binding protein from the triatomine *Rhodnius prolixus*[41].

In contrast, we have demonstrated the mechanism by which CxD7L1 inhibits platelet aggregation, never reported before in the D7 protein family. CxD7L1 inhibited aggregation induced by low doses of ADP or collagen in a dose-dependent manner. Platelet aggregation induced by low doses of collagen is known to be highly dependent on ADP release from platelet granules, as platelets treated with apyrase or ADP receptor antagonists poorly respond to these agonists[42,43]. CxD7L1 showed an inhibitory effect on aggregation triggered by the $TXA_2$ pathway, as it attenuated aggregation induced by both U-46619 and arachidonic acid, the $TXA_2$ precursor, which suggests that CxD7L1 interacts with $TXA_2$. However, we showed CxD7L1 does not bind $TXA_2$ through ITC and aggregation studies, ruling out the direct interaction between CxD7L1 and $TXA_2$. It is known that aggregation through $TXA_2$ is linked to ADP signaling[44]. This observation agrees with a previous description of a *R. prolixus* aggregation inhibitor 1 (RPAI-1) which binds ADP and interferes with $TXA_2$ pathways[28]. Taken all together, we demonstrated that CxD7L1 inhibits platelet aggregation by sequestering ADP, which is released from platelet dense granules upon platelet activation promoting a stable platelet response[37,38,45]. By removing secreted ADP from the vicinity of the platelet, CxD7L1 prevents ADP from performing its role of platelet propagation.

Adenine nucleotides and derivatives play an important role in vascular biology and immunology at the mosquito bite site. ATP and ADP induce constriction of blood vessels and ADP acts as a potent mediator of platelet aggregation in mammals. Metabolism of ATP and ADP would lead to the production of AMP by apyrases that would be further metabolized to adenosine by 5-nucleotidase. Apyrases have been found in the saliva of most blood-feeding arthropods studied so far[12]. The ability of CxD7L1 to scavenge ATP and ADP may compensate for the low salivary apyrase activity detected in *C. quinquefasciatus* compared to *Ae. aegypti*[46]. CxD7L1 also binds and scavenges adenosine. Although adenosine causes vasodilation and inhibits platelet aggregation, it triggers pain and itch responses by inducing mast cell degranulation. Pain and itch may alert the host to the presence of a biting mosquito, preventing a successful blood meal[47].

Rabbit anti-CxD7L1 IgG and anti-CxD7L2 IgG did not prevent D7 proteins from binding their ligands ADP and serotonin, respectively (Supplementary Fig. 9). Therefore, blocking the proteins' activities with antibodies was not a feasible approach to investigate their role in blood feeding in vivo, as previously done with other insect salivary proteins[48]. We demonstrated the ability of *Culex* mosquito bites to scavenge ADP at the bite site, effect attributed to CxD7L1, and to inhibit host hemostasis in vivo using the tail bleeding assay. This assay is widely used as an in vivo assessment of the hemostatic action of platelets in rodents and is considered a general approach to assess primary hemostasis. Bleeding time is mainly determined by the interaction between platelets and damaged vessel wall leading to the hemostatic plug formation[49,50]. The prolonged bleeding time we observed in mice whose tails were immersed in CxD7L1 matched the research of other authors who linked the presence of an ADP platelet receptor deficiency with extended bleeding time in mice[51].

Arthropods underwent multiple independent evolutionary events to adapt to consume blood meals from different or new hosts. This independent evolutionary scenario has led to a great variety of salivary protein families that have acquired different functions related to blood-feeding. Gene duplication is an important mechanism for the evolution of salivary proteins. Duplication of D7 genes may have been advantageous in providing greater amounts of D7 proteins at the bite site to counteract high concentrations of host mediators[52]. Gene duplication combined with the pressure of the host hemostatic and immune responses may have led to functional divergence as observed in the D7 short proteins from *An. gambiae* and their specialization towards different biogenic amines[10]. The D7 protein family is polygenic in all Nematocera so far studied[53]. In *C. quinquefasciatus*, D7 genes are also a result of gene duplication events, given the number of genes that encode D7 proteins and their location in the genome on chromosome 3[54]. *C. quinquefasciatus* mosquitoes are traditionally considered bird-feeders that later adapted to mammalian blood-feeding. They are increasingly recognized as important bridge vectors, vectors that acquire a pathogen from an infected wild animal and subsequently transmit the agent to a human, based on studies that examine host preference, vector/host abundance, viral infection rates, and vector competence[55]. *C. quinquefasciatus* contain potent salivary proteins that counteract bird thrombocytes aggregation mediators such as serotonin and platelet activation factor (PAF). We have demonstrated that CxD7L2 tightly binds serotonin while Ribeiro et al. demonstrated that PAF phosphorylcholine-hydrolase inhibits PAF enzymatically[56]. Thrombocytes are not responsive to ADP[57,58], but ADP is an important mediator of platelet aggregation in mammals. We hypothesize that the function of ADP-binding by CxD7L1 protein has arisen from the selective pressure of mammalian hemostatic responses. This acquired D7-ADP-binding function may have provided an advantageous trait in *C. quinquefasciatus* mosquitoes that helped them to adapt to blood-feeding on mammals. *C. tarsalis* mosquitoes prefer to feed on birds but will readily feed on mammals in the absence of their preferred host[59]. An alignment between CxD7L1 and *C. tarsalis* D7 long proteins showed that most of the residues involved in ADP binding are conserved in *C. tarsalis*, suggesting that D7 proteins that bind ADP may be widespread in the genera *Culex*. More studies are necessary to confirm this hypothesis.

In conclusion, we determined the binding capabilities of the CxD7L1 and CxD7L2 proteins and demonstrated their role in inhibiting human platelet aggregation through different mechanisms of action ex vivo and preventing blood hemostasis in vivo. We identified a function of ADP-binding in the well-characterized D7 protein family. Moreover, the structure of the complex CxD7L1-ADP was solved, showing a different binding mechanism for a D7 with the binding pocket located between the N-terminal and C-terminal domains whereas most D7s bind ligands within one of these two respective domains. These proteins help blood feeding in mosquitoes by scavenging host molecules at the bite site that promote vasoconstriction, platelet aggregation, itch, and pain. Accumulation of these proteins in the salivary glands of females confers an evolutionary advantage for mosquito blood feeding on mammals.

## Methods

**Ethics statement**. Public Health Service Animal Welfare Assurance #A4149-01 guidelines were followed according to the National Institute of Allergy and Infectious Diseases (NIAID), National Institutes of Health (NIH) Animal Office of Animal Care and Use (OACU). These studies were carried out according to the NIAID-NIH animal study protocols (ASP) approved by the NIH Office of Animal Care and Use Committee (OACUC), with approvals ID ASP-LMVR3 and ASP-LMVR102. Mice used in this study were housed in one of the animal facilities from the NIAID/NIH and were humanely treated according to OACU regulations.

**Mosquito rearing, salivary gland and saliva collection**. *C. quinquefasciatus* mosquitoes were reared in standard insectary conditions at the Laboratory of Malaria and Vector Research, NIAID, NIH (27 °C, 80% humidity, with a 12-h light/dark cycle) under the expert supervision of Andre Laughinghouse, Kevin Lee, and Yonas Gebremicale. The mosquito colony was initiated from egg rafts collected in Hilo, HI, USA, and maintained at NIH since 2015. Salivary glands from sugar-fed 4–7-day-old female mosquitoes were dissected in phosphate-buffered saline

(PBS), pH 7.4 using a stereomicroscope. SGE was obtained by disrupting the gland wall by sonication (Branson Sonifier 450). Tubes were centrifuged at $12,000 \times g$ for 5 min and supernatants were kept at $-80\,°C$ until use. Oil-induced saliva was collected as previously described[60]. Briefly, alive mosquitoes were immobilized by placing their back on sticky tape. Mosquito mouthparts were inserted into 10 μL pipette tips containing mineral oil and salivation was promoted by injection of 200 nL of 3.6 mg/mL pilocarpine intrathoracically. 1 h after salivation, pipette tips that contained oil with saliva droplets were combined in a tube with 10 μL of PBS and aqueous phase was separated by centrifugation.

**CxD7L1 and CxD7L2 gene expression pattern**. *C. quinquefasciatus* larvae (stages L1–L4 categorized by age and size), pupae, and 0–2-day-old adults (male and female) were collected and kept in Trizol reagent (Life Technologies). Additionally, female adults were dissected, head and thorax were separated from abdomens, and independently analyzed. In all cases each sample consisted of 10 specimens. Total RNA was isolated with Trizol reagent following the manufacturer instructions (Life Technologies). cDNA was obtained with the QuantiTect Reverse Transcriptase Kit (Qiagen), from 1 μg of starting RNA. Nanodrop ND-1000 spectrophotometer was used to determine all concentrations and $OD_{260/280}$ ratios of nucleic acids. qPCR was carried out as previously described[61]. Specific primers to target CxD7L1 and CxD7L2 genes were designed (CxD7L1-F: 5′-ACGGAAGCATGGTTTTTCAG-3′, CxD7L1-R: 5′-GGATTGCAGATTCGTCCATT-3′, CxD7L2-F: 5′-CCACGAACAA CAACCATCTG-3′, CxD7L2-R: 5′-CACGCTTGATTTCATCAGGA-3′). Briefly, in a final volume of 20 μL, reaction mix was prepared with 2X SsoAdvanced Universal SYBR Green Supermix (Bio-Rad), 300 nM of each primer, and 100 ng of cDNA template. Two biological replicates were tested. All samples were analyzed in technical duplicates and non-template controls were included in all qPCR experiments as negative controls. qPCR data were manually examined and analyzed by the ΔΔCt method using the CFX Maestro software version 1.1 (BioRad). ΔCt values were obtained by normalizing the data against *C. quinquefasciatus* 40S ribosomal protein S7 transcript (AF272670; CxS7-F: 5′-GTGATCAAGTCCGG CGGTGC-3′ and CxS7-R: 5′-GCTTCAGGTCCGAGTTCATCTC-3′) as the reference gene. Male adult samples were chosen as controls for the ΔΔCt values. Relative abundance of genes of interest was calculated as $2^{-\Delta\Delta Ct}$. Graphs were prepared using GraphPad Prism software version 8.02.

**Cloning, expression, and purification of recombinant proteins**. CxD7L1 and CxD7L2 coding DNA sequences (AF420269 and AF420270) were codon-optimized for mammalian expression and synthesized by BioBasic Inc. VR2001-TOPO vectors containing CxD7L1 and CxD7L2 sequences (Vical Incorporated) and a 6x-histidine tag were transformed in One Shot TOP10 chemically competent *E. coli* (Invitrogen). FreeStyle 293-F human embryonic kidney cells (ATCC; Cat. no.: CRL-10852) were transfected with sterile plasmid DNA, prepared with EndoFree plasmid MEGA prep kit (Qiagen, Valencia, CA), at the SAIC Advance Research Facility (Frederick, MD), and supernatants were collected 72 h after transfection. Recombinant proteins were purified by affinity chromatography followed by size-exclusion chromatography, using Nickel-charged HiTrap Chelating HP and Superdex 200 10/300 GL columns, respectively[62]. Briefly, supernatants containing the secreted recombinant protein were concentrated from 1 L down to approximately 300 mL using an Amicon stirred cell with a cellulose membrane of 10 kDa nominal molecular weight (Millipore Sigma). The concentrated sample was supplemented with 500 mM NaCl and 5 mM imidazole and was applied to a Nickel-charged HiTrap Chelating HP column using a peristaltic pump. Then, the column was washed with 2 column volumes of 25 mM phosphate buffer, 500 mM NaCl, 5 mM imidazole, pH 7.4. The recombinant protein was eluted by passing a gradient of 25 mM phosphate buffer, 500 mM NaCl, 1 M imidazole, pH 7.4 using an AKTA high-performance liquid chromatography system (GE Healthcare). The protein fractions were visualized in a NuPage gel electrophoresis. The fractions of interest were pooled and concentrated using Amicon® Ultra-15 centrifugal filter units (Millipore Sigma) and used for the subsequent size exclusion chromatography with a Superdex 200 10/300 GL column.

To determine the crystal structure, recombinant CxD7L1 was produced in *E. coli*. The CxD7L1 coding DNA sequence was amplified by PCR from cDNA of *C. quinquefasciatus* salivary glands and was cloned in pET-17b plasmid and expressed in BL21 pLysS cells (Invitrogen). Protein expression was carried out as previously described[62]. The bacterial strains were grown at 37 °C in LB broth and induced with 1 mM isopropyl 1-thio-β-D-galactopyranoside for 3 h at 250 rpm. Inclusion bodies were harvested, washed, and solubilized by mixing the bacterial pellet with 20 mL of 6 M guanidine hydrochloride, 25 mM Tris–HCl, 1 mM EDTA for 1 h at room temperature. Dithiothreitol at a final concentration of 10 mM was added to the solution. The protein was refolded by adding the guanidine-solubilized protein solution dropwise into 4 L of 200 mM arginine, 50 mM Tris, 1 mM reduced glutathione, 0.2 mM oxidized glutathione, 1 mM EDTA, pH 8.0 and incubating with stirring at 4 °C overnight. Bacterial CxD7L1 was concentrated first by tangential flow filtration (from 4 L to ~400 mL) and then by an Amicon stirred cell filtration unit, with a cellulose membrane of 10 kDa nominal molecular weight (Millipore Sigma). The concentrated sample was purified by size exclusion chromatography, using a HiPrep 16/60 Sephacryl S-100 HR column and PBS as elution buffer. The protein fractions of interest were dialyzed against 4 L of 25 mM 2-(N-morpholino)ethanesulfonic acid (MES), pH 6.0 overnight at 4 °C with gentle

stirring. It was followed by cation exchange chromatography with a HiPrep SP FF 16/10 column with an elution step with a gradient of 25 mM, 1 M MES, pH 6.0. A last step of analytical size exclusion chromatography was performed using a Superdex 200 10/300 GL column with 25 mM Tris, 50 mM NaCl, pH 7.4. All high-performance liquid chromatography (HPLC) analyses were performed using Unicorn software version 5.3.1 (GE Healthcare).

To determine the binding capacity of the N-terminal and C-terminal domains of CxD7L2, we independently cloned the correspondent cDNA into pCR2.1 TOPO vector (Invitrogen). CxD7L2-NT nucleotides 1–150 and CxD7L2-CT nucleotides 151–293 were amplified by PCR and subcloned into pCR2.1 After verifying the sequence identity by sequencing, the cDNA was subcloned into pET-17b plasmid between NdeI and XhoI restriction sites. Protein expression was carried out using BL21 pLysS cells (Invitrogen). N-terminal and C-terminal domains were purified as described above. All HPLC columns were obtained from GE Healthcare Life Science, Piscataway, NJ. All purified proteins were separated in a 4–20% NuPAGE Tris-glycine polyacrylamide gel and visualized by Coomassie stain. Protein identity was verified by Edman degradation at the Research Technologies Branch, NIAID, NIH.

**Polyclonal antibody production**. Polyclonal antibodies against CxD7L1 and CxD7L2 were raised in rabbits. Immunization of rabbits was carried out in Noble Life Science facility (Woodbine, MD) according to their standard protocol. Briefly, rabbits received a total of four immunizations with 1 μg of recombinant protein each at days 0, 21, 42, and 63. Freund's Complete Adjuvant was used for the initial injection and Freund's Incomplete Adjuvant was used for subsequent injections. Rabbit sera were shipped to our laboratory where purification of IgG was performed by affinity chromatography using a 5-mL HiTrap protein A HP column following manufacturer's instructions (GE Healthcare, Piscataway, NJ). Purified IgG protein concentration was determined by Nanodrop ND-1000 spectrophotometer. Additionally, antibodies against *C. quinquefasciatus* SGE were raised in rabbits. Levels of specific antibodies were determined by enzyme-linked immunosorbent assay (ELISA) according to Chagas et al.[33]. Briefly, microtiter flat bottom plates (Maxisorp, Nunc, Roskilde, Denmark) were coated with 100 ng of recombinant proteins per well in carbonate bicarbonate buffer, pH 9.5 (Sigma) at 4 °C overnight. Plates were blocked with 5% bovine serum albumin (BSA) in Tris-buffered saline (TBS) (25 mM Tris, 150 mM NaCl, pH 7.4) for 2 h at room temperature. After three washes with TBS supplemented with 0.05% (v/v) Tween (TTBS) primary rabbit antibodies diluted at 1:1000 in TTBS were added. After 1 h incubation and further washing, alkaline phosphatase-coupled anti-rabbit IgG (1:10,000 in TTBS, Sigma cat. no.: A3687) was added. Following another washing cycle, plates were developed with stabilized p-nitrophenyl phosphate (Sigma) and absorbance was measured at 405 nm in a VersaMax microplate reader (Molecular Devices) after a 15-min incubation.

**Protein quantification by ELISA**. The amount of CxD7L1 an CxD7L2 in the salivary glands of *C. quinquefasciatus* was assessed by ELISA, as previously described[33,48]. Plates were coated with a serial dilution of recombinant proteins (2000–0.98 ng) to generate a standard curve (sigmoidal fit, $R^2 = 0.9909$) which was used to infer the amount of D7 proteins present in *C. quinquefasciatus* SGE obtained from individual mosquitoes dissected before ($n = 15$) and after blood-feeding on chicken ($n = 15$). Samples from oil-induced saliva collected from five mosquitoes ($n = 5$) were also included. To specifically quantify each D7 protein, rabbit IgG antibodies were pre-adsorbed to prevent that cross-reactivity between CxD7L1 and CxD7L2 antibodies would mask the quantification results. Briefly, 50 μg of each recombinant protein was deposited onto a nitrocellulose membrane and blocked for 1 h with Tris–HCl, pH 7.4, 150 mM NaCl (TBS) containing 5% (w/v) powdered non-fat milk (blocking buffer). Then, the membrane containing CxD7L1 was incubated overnight with anti-CxD7L2 IgG diluted to 0.1 μg/mL in 0.05% Tween, TBS and CxD7L2 membrane was incubated with anti-CxD7L1 antibodies. These pre-adsorbed antibodies were used for the ELISA. After 1 h incubation and further washing, alkaline phosphatase-coupled anti-rabbit IgG (1:10,000 in TTBS, Sigma cat. no.: A3687) was added. Absorbance was measured at 405 nm in a VersaMax microplate reader (Molecular Devices) during 1 h after the addition of the substrate (p-nitrophenyl phosphate, Sigma) and readouts at 30-min were used for the fitting analysis (15, 30, and 45 min incubation lead to similar results).

**Western blot**. *C. quinquefasciatus* SGEs (2.5 μg) and 100 ng of CxD7L1 and CxD7L2 were separated by NuPAGE. Proteins were transferred to a nitrocellulose membrane (iBlot, Invitrogen) that was blocked overnight at 4 °C with blocking buffer. Purified and pre-adsorbed anti-CxD7L1 and anti-CxD7L2 IgG antibodies were diluted in blocking buffer (0.5 μg/mL) and incubated for 90 min. Goat anti-rabbit conjugated to alkaline phosphatase (Sigma cat. no.: A3687) diluted in blocking buffer (1:10,000) was used as a secondary antibody and immunogenic bands were developed by the addition of BCIP/NBT substrate (Promega). The reaction was stopped with distilled water.

**Immunolocalization of CxD7L1 and CxD7L2**. *C. quinquefasciatus* salivary glands were dissected in PBS, transferred to a welled plate, and fixed with 4% paraformaldehyde (Sigma) for 30 min at room temperature. Tissues were washed

3 times for 10 min each with 1× PBS to remove paraformaldehyde and then blocked with 2% BSA, 0.5% Triton X-100, 1× PBS, pH 7.4 overnight at 4 °C. Glands were washed 3 times with PBS to remove Triton X-100 and were transferred to clean wells to which 200 μL of 1 μg/mL pre-adsorbed antibodies against either CxD7L1 or CxD7L2 (raised in rabbits and diluted 1:1000 in 2% BSA 1× PBS) were added. Glands incubated in 2% BSA 1× PBS served as a negative control. Plate wells were covered and incubated overnight at 4 °C. Primary antibodies were removed by 3 washes with 2% BSA 1× PBS and incubated with 2 μg/mL anti-rabbit IgG Alexa Fluor 594 (Thermo Fisher cat. num.: A11012) for 2 h in the dark at 4 °C. Conjugate was removed by 3 additional washes with 1× PBS. DNA was stained with 1 μg/mL DAPI (Sigma D9542) and actin with 0.04 μg/mL Phalloidin Alexa 488 (Invitrogen cat. num.: A12379) for 20 min. Glands were washed 3 times with PBS and transferred to glass slides containing droplets of PBS. PBS was removed without drying the glands, and tissues were mounted using a coverslip coated with 25 μL Prolong Gold mounting medium. Slides were covered and left to dry at room temperature and then stored at 4 °C. Bright field and fluorescent images were acquired in a Leica Confocal SP8 microscope with a 63× objective using Navigator tool. Images were processed with Imaris software version 9.2.1 and postprocessing was carried out in Fiji ImageJ version 1.52n for representative purposes.

**Isothermal titration calorimetry (ITC).** Thermodynamic binding parameters of CxD7L1 and CxD7L2 to several pro-hemostatic ligands were tested using a Microcal VP-ITC microcalorimeter. The panel of substances tested included several nucleosides/nucleotides or derivates (ATP, ADP, 5′-AMP, 3′-AMP, cyclic AMP, adenosine, GTP, TTP, inosine, sodium polyphosphate, Sigma-Aldrich), biogenic amines (epinephrine, norepinephrine, histamine, serotonin, Sigma-Aldrich), and pro-inflammatory/pro-hemostatic lipid compounds ($LTB_4$, $LTC_4$, $LTD_4$, $LTE_4$, arachidonic acid, and the stable analog of $TXA_2$: U-46619, Cayman Chemicals). Ligands and protein solutions were prepared in 20 mM Tris–HCl, pH 7.4, 150 mM NaCl (TBS) at 30 and 3 μM, respectively. Lipid ligands were prepared by evaporating the ethanol or chloroform solvent to dryness under a stream of nitrogen. Lipid ligands were further dissolved in TBS and sonicated for 10 min (Branson 1510) to ensure dissolution. Lipid ligands were used at 50 μM of ligand and 5 μM of protein. Injections of 10 μL of ligand were added to the protein samples contained in the calorimeter cell at 300 s intervals. Experiments were run at 30 °C. Thermodynamic parameters were obtained by fitting the data to a single-site binding model in the Microcal Origin software package version 7 (OriginLab). For saturation studies, CxD7L2 protein was pre-incubated with 50 μM serotonin for 30 min and titrated with $LTD_4$.

**CxD7L1 crystallization and structure determination.** Purified protein was incubated overnight at 4 °C with 1.2 times molar excess of ADP. Crystals were obtained using the hanging drop-vapor diffusion method with 0.01 M zinc sulfate heptahydrate, 0.1 M MES monohydrate, pH 6.5, and 25% v/v polyethylene glycol monomethyl ether 550 (Crystal Screen 2, Condition 27, Hampton Research).

For data collection, the crystals were rapidly soaked in the mother liquor solution (the crystallization buffer described above) supplemented with 25% glycerol and flash frozen in a nitrogen gas stream at 95 K. Data were collected at beamline 22BM at the Advanced Photon Source, Argonne National Laboratory equipped with 10 Hz Rayonix MX300HS detector. A crystal that diffracted to 1.97 Å resolution with cell dimensions (in Å) of $a = 76.66$, $b = 84.32$, and $c = 132.07$ and belonged to the orthorhombic space group $I2_12_12_1$ (Table 3) was used to collect a data set. The data were processed, reduced and scaled with XDS[63]. The structure of CxD7L1 was determined by molecular replacement using Phaser version 2.8.3[25] by employing separate, manually constructed search models for the N-terminal and C-domains based on the crystal structure of *An. stephensi* AnStD7L1 (PDB ID: 3NHT). The final model of CxD7L1 was constructed by iterative manual tracing of the chain using the program Coot version 0.8.9.2[64] after each cycle of refinement with stepwise increase in the resolution using Phenix version 1.17.1-3660. All structural figures were produced with PyMOL (PyMOL molecular graphics system, version 1.7.4; Schrödinger, LLC) and UCSF Chimera version 1.13.1 (Resource for Biocomputing, Visualization, and Informatics at the University of California, San Francisco)[65].

**Platelet aggregation assay.** Platelet-rich plasma (PRP) was obtained from normal healthy donors on the National Cancer Institute Institutional Review Board (NCI IRB) approved NIH protocol 99-CC-0168, "Collection and Distribution of Blood Components from Healthy Donors for In Vitro Research Use." Research blood donors provide written informed consent, and platelets were de-identified prior to distribution. Platelet aggregation was measured using an aggregometer (Chrono-Log Corporation). Briefly, 300 μL of PRP, diluted 1:3 to approximately 250,000 platelets/μL in Hepes-Tyrode's buffer (137 mM NaCl, 27 mM KCl, 12 mM NaHCO₃, 0.34 mM sodium phosphate monobasic, 1 mM MgCl₂, 2.9 mM KCl, 5 mM Hepes, 5 mM glucose, 1% BSA, 0.03 mM EDTA, pH 7.4) were pre-stirred in the aggregometer for 1 min to monitor pre-aggregation effects. Different concentrations of recombinant proteins or TBS as negative control were added to the PRP before adding the agonists. Aggregation agonists used in our studies included native collagen type I fibrils from equine tendons, convulxin, ADP, U-46619, arachidonic acid, serotonin, epinephrine, or combination of agonists. Their

concentrations are specified in the figure captions. Technical duplicates were performed. Graphs were prepared using GraphPad Prism software version 8.02.

**ADP determination.** ADP levels were determined using the ADP fluorometric assay kit (Abcam; cat. no.: ab211087), based on the rapid conversion of ADP to an intermediate form by an ADP enzyme mix provided in the kit which reacts with a fluorescent probe to generate a strong stable fluorescent signal. Sixteen to twenty-week-old female mice (Balb/c) were anesthetized using a mixture of ketamine and xylazine (90 mg/kg, 10 mg/kg IP). Their right ears were exposed for 10 min to 5–7-day-old mosquitoes that had starved overnight while their left ears were protected from bites and remained unbitten. Immediately after feeding, mice were euthanized and biopsies were taken from bitten and non-bitten ear skin using a 3 mm biopsy punch (Acu-Punch, Acuderm). Tissues were washed with 500 μL of cold PBS and homogenized in 200 μL of the kit buffer containing 1× Halt protease inhibitor (Thermo Scientific) with a pestle during 2 min. Homogenized samples were incubated in ice for 10 min and clarified by centrifugation for 5 min, 10,000 g at 4 °C. Supernatant was transferred to a 10 kDa Amicon ultra centrifugal filter (Millipore) and centrifuged for 10 min at 12,000 g at 4 °C to remove the ADP bound to a salivary molecule (larger than 10 kDa). Tests were performed following the manufacturer's recommendations in biological duplicates and technical duplicates. Signal at Ex/Em = 535/587 was read using a Cytation 5 image reader (Biotek) and results were inferred from an ADP standard curve using a linear regression analysis ($R^2 = 0.9932$). Samples of known ADP concentration (1 μM) in the presence of absence of equimolar concentrations of CxD7L1 were also analyzed. Graphs were prepared using GraphPad Prism software version 8.02.

**Tail vein bleeding assay.** Balb/c mice were anesthetized and placed prone on a warming pad at 37 °C. The tail was measured and marked with ink at 1 cm from the tip and transected with a scalpel to lacerate the tail vein. The tail was hung over the edge of the warming pad and immersed in warmed tubes at 37 °C containing 200 μL of either saline as a control or D7 recombinant proteins at three different concentrations (2.5, 5, and 10 μM) ensuring that the tail tip did not touch the walls or the bottom of the tube. The time from the incision to the cessation of bleeding was recorded as the tail vein bleeding time. Graphs were prepared using GraphPad Prism software version 8.02.

**CxD7L3 bioinformatic, qPCR, and mass spectrometry studies.** Two sets of specific primers located near the unique nucleotide insertion characteristic of CxD7L3 (AY388553) were designed: CxD7L3_a For: 5′-CTAACTCAATCCATCA CGATGC-3′, CxD7L3_a Rev: 5′-CAGTTGTTCAGAGCGTTGTCC-3′ and CxD7L3_b For: 5′-GTCTGAGATATTTCGATAGAGATGG-3′, CxD7L3_b Rev: 5′-GTTTGACTTCAGCAGGCTGC-3′. gDNA was extracted from female and male adult mosquitoes using the Phire Animal Tissue Direct PCR Kit, and PCR was performed following the manufacturer's recommendations. *C. quinquefasciatus* CxD7L3 gene expression analysis was done as described in the "Methods" section CxD7L1 and CxD7L2 gene expression pattern using the 40S ribosomal protein S7 as the housekeeping gene. As samples, we used cDNA from head and thorax of female adult mosquitoes and gDNA extracted from female and male adults. Graphs were prepared using GraphPad Prism software version 8.02.

In order to determine whether the unique insertion of 20 amino acids characteristic of CxD7L3 was present in the SGE of *C. quinquefasciatus*, we performed mass spectrometry analysis of three independently dissected salivary glands sets. Samples were submitted to liquid chromatography coupled with mass spectrometry at the Research and Technology Branch (NIAID, NIH). Samples were reduced with 5 mM DTT for 40 min at 37 °C, cooled to room temperature and alkylated with 15 mM iodoacetamide for 20 min. Then, extracts were hydrolyzed with 200 ng of trypsin (37 °C for 15 h in a final volume of 40 μL). The solution was evaporated to near dryness under vacuum at 50 °C. Twenty-five microliters of 0.1% trifluoroacetic acid (TFA) was added and the pH was adjusted to 2.5 with the addition of 10% TFA. Samples were desalted with C18 OMIX 10 solid phase extraction tips. The digests were eluted with 0.1% TFA, 50% acetonitrile, and dried under vacuum. The peptides were dissolved in 12 μL of 0.1% formic acid, 3% acetonitrile which was used as the injection solvent. Digested peptides were subjected to the LC–MS analysis using Orbitrap Fusion mass spectrometer (ThermoFisher Scientific) connected with EASY nLC 1000 liquid chromatography system. Nano-LC was carried out with a 5 μL injection onto a PepMap 100 C18 3-μm trap column (2 cm, ID 75 μm) and a 2 μm PepMap RSLC C18 column from (25 cm, ID 75 μm), both from ThermoFisher Scientific. The LC was operated at a 300 μL/min flow rate with a 100-min linear gradient from 100% solvent A (0.1% formic acid and 99.9 % water) to 40% solvent B (0.1% formic acid, 20% water, and 79.9% acetonitrile) followed by a column wash. A standard data-dependent acquisition was performed with a full MS spectrum is obtained by the Orbitrap for m/z 400–2000 at the resolution of 120,000 with EASY-IC calibration. The precursor ions, with charges from 2 to 8, were selected, isolated (1.6 m/z window), fragmented by CID, then scanned by the Ion Trap. Survey scans were performed every 2 s and the dynamic exclusion was enabled for 30 s. Acquisitions were searched against the NCBInr proteome and the cRAP.fasta database (theGPM.org) using PEAKS v10 (Bioinformatics Solutions Inc., Ontario, Canada) and a semi-tryptic search strategy with tolerances of 6 ppm for MS and 0.5 Da for MS/MS and

carbamidomethylation of cysteine as a fixed modification and oxidation of methionine as a dynamic modification allowing for two missed cleavages. Peptides were filtered with a 0.5% false discovery rate using a decoy database approach and a two spectral matches/peptide requirement.

**Reporting summary**. Further information on research design is available in the Nature Research Reporting Summary linked to this article.

## Data availability

The source data underlying Figs. 1a–c, 6a–c, and 7a, b and Supplementary Figs. 1c, 2b, 3b, c, 4c, 5a, 6, 7a–d, and 8 are provided as a Source Data file. The datasets generated and/or analyzed during the current study are available from the corresponding author on reasonable request. The coordinates and structure factors of the crystal structure of CxD7L1 in complex with ADP have been deposited in the Protein Data Bank under the accession number PDB 6V4C. Databases used in the study are as follows: NCBInr proteome (available at www.ncbi.nlm.nih.gov), the common Repository of Adventitious Proteins: cRAP.fasta database (available at ftp://ftp.thegpm.org/fasta/cRAP), PDB (https://www.rcsb.org/). All other data are available from the corresponding author on reasonable request. Source data are provided with this paper.

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

## Acknowledgements

The authors thank Kevin Lee, Andre Laughinghouse, and Yonas Gebremicale for excellent mosquito rearing and Van My Pham for salivary glands dissection. The authors also thank John Andersen and Jose Ribeiro for relevant scientific discussion, Thrity Avary from Chrono-Log Corporation for technical assistance with platelet aggregation studies. The authors thank Glenn Nardone and Lisa (Renne) Olano, Research Technology Branch, NIH, for the mass spectrometry analysis and Bradley Otterson, NIH Library Writing Center, for manuscript editing assistance. This research was supported by the Intramural Research Program of the NIH/NIAID (AI001246-01).

## Author contributions

Planned experiments: I.M.-M. and E.C. Performed experiments: I.M.-M., A.P., P.C.V.L., A.G.G., O.K., B.B., A.C.C., S.G., and E.C. Analyzed data: I.M.-M., A.G.G., L.B.S., D.N.G., and E.C. Wrote the paper: I.M.-M. and E.C.

## Competing interests

The authors declare no competing interests.
