## [Peer Review File · Nature Communications]

Reviewers' comments:

Reviewer #1 (Remarks to the Author):

A. Summary of the key results

The investigators screened a number of possible ligands for binding to two long form D7 proteins and found a new putative binding partner – adenine nucleotides and nucleosides, including ADP. A crystal structure is presented that reveals the binding pocket for ADP. It is noted that ADP binding may play a role in limiting hemostasis since ADP plays a role in platelet aggregation. Biochemical assays are presented showing that one D7 protein can inhibit platelet aggregation when low doses of ADP are used as an agonist.

B. Originality and significance

A number of proteins and functions have been ascribed to mosquito saliva that prevent hemostasis and facilitate blood feeding. The mechanism described here is new, but does not add much to the literature since it is well established that mosquito saliva inhibits platelet aggregation. This is a new mechanism, but it is not clear how important this mechanism is in vivo since only one form of D7 utilizes this mechanism, this mechanism only works with low doses of ADP, and there are many other agonists present in vivo that would induce platelet aggregation. D7 was able to inhibit platelet aggregation in vitro when ADP was used as an agonist, but it is not clear how important this is or if it would work in vivo. It is also not clear how important this mechanism is in individuals who are atopic and who release much higher levels of histamine and other biogenic amines, which would saturate the D7 proteins. It is possible that the impact of D7 proteins is limited to non-sensitive individuals.

C. Data and methodology

The data presented in the manuscript appear thorough and sophisticated, although it is an in vitro analysis with recombinant proteins, and it is not known how important this new mechanism is in vivo.

D. Appropriate use of statistics

The statistics are fine.

E. Conclusions

The study is very well performed and the conclusions are well reasoned based on the data that is presented. The impact is lessened because this is an entirely in vitro and biochemical analysis of a protein and putative ligand, and it is unclear how important this mechanism is in vivo. This is especially important since there are many redundant mechanisms in mosquito saliva that serve to prevent hemostasis and promote blood feeding.

F. Suggested improvements

Negative controls would be helpful in Table 1 to show that the assay was properly calibrated. It would also be helpful to determine how many D7 molecules are expectorated into skin and how many ADP molecules are expected to be present in a mosquito bite lesion in non-sensitive and hypersensitive individuals. This is critical to estimate the feasibility of D7 in scavenging enough ADP to have a physiologically relevant impact, and absolutely necessary in the absence of an in vivo experiment. This is lightly addressed in lines 382-386 but some experimental evidence would be helpful to determine if there really is enough D7 present in an inoculum to bind enough of this material.

G. References

The references are fine.

H. Clarity and context

The manuscript is very well written and easy to follow the text and figures. Well done.

Reviewer #2 (Remarks to the Author):

This is a complex study combining biochemical studies, crystallography and ex-vivo experiments. Authors demonstrated different ligand-binding affinities of two *Culex quinquefasciatus* D7 proteins and revealed a novel binding affinity for adenine nucleotides/nucleosides in CxD7L1 D7 protein. Introduction is comprehensive and well written, methods are written with enough details. Data acquired may prompt researchers in the field to reveal similar affinities in other *Culex* species or to enlarge current knowledge of long D7 proteins in *Anopheles* and *Aedes* mosquitoes. Results are interesting and novel, nevertheless I like to ask authors to clarify/answer some important details.

1. Did you confirm by ITC that CxD7L2 do not bind any nucleotides/nucleosides?
2. Could you estimate what amount of D7 proteins of interest is injected into the feeding site by *Culex quinquefasciatus* mosquito? Is the final concentration of these proteins high enough to sequester ligands presented at the bite site at 1-10 μM concentration?
3. What is your opinion about the third D7 long form CQ_LD7_3, a protein similar to CxD7L2 (73% identity)? Did you rule out the possibility of cross-reaction between these two proteins in your immunolocalization experiment?
4. Discussion is well-written, I agree with the hypotheses that this D7 protein partially substitutes less active apyrase in *Culex*. However, I would greet any supporting data that this novel ADP-binding function evolved to enhance bloodfeeding in mammals. It would be useful to compare binding activities of D7 proteins in closely related species *Culex pipiens pipiens* which is ornithophilic. I believe that such comparison would be more informative than the comparison with *C. tarsalis* which is opportunistic feeder and more distant phylogenetically from *C. quinquefasciatus*.

Specific comments to figures

1. Alignment shown in Fig. 1 includes only mosquito D7 proteins despite sand fly D7s are mentioned in the preceding text. Why you did not include, for example, *P. duboscqi* D7-related protein (PDB ID: 6MT7)?
2. Residues involved in ADP binding are not marked in the Fig. 1. On contrary, they are highlighted in blue in Suppl. Fig. 3. These highlighted amino acids present in *C. tarsalis* D7 homologs can be found also in AeD7 (both Y conserved) and 3AnSTD7L1 (conserved Y, K and Y) in Fig 1. I suggest mentioning this in the text and highlighting the amino acids in in Fig. 1.
3. In Fig. 4, result for LTB4 is not presented (in contrast to LTD4 or leukotrienes with no binding affinities). Please add the ITC curve for LTB4 into Fig. 4 (or mention in the text on line 190 that the data are not shown).

4. Could you mark all amino acids mentioned in the text following paragraph (including K146, N265, R271 and S263) in the figure 5e and point to the exact amino acid by an arrow? Missing arrows makes the figure difficult to follow.

Minor comments or typos

Lines 17-18: Please modify “mosquitoes” to “mosquito” (or “vector” to “vectors”).

Line 35: Add a comma after the bracket.

Line 38: Please modify “They can also” to “It also can”.

Table 1: consider to divide this table into two individual ones, each being close to the relevant ITC figure.

Line 166: Modify “j-f” to “f-j”.

Line 200: This part is not mentioned in the methods. Please include short methodology into the section 4.4.

Line 201: Please correct “CxD7L1-CT” to “CxD7L2-CT”.

Line 219: Modify “C-domains” to “C-terminal domains”.

Line 260: I do not understand why you refer to Suppl. Fig. 3 here (the figure shows the sequence similarity with *Culex tarsalis* D7 proteins).

Line 321: Correct “Fig. 4h” to “Fig. 4i”.

Line 356: Please include the concentrations of the recombinant proteins into Fig. 9.

Line 370: Modify “expressed” to “localized” (protein was detected by immunolocalization).

Lines 485-489: Please find more appropriate reference to support your statement. Study in reference 46 describes platelet-activating-factor-hydrolyzing phospholipase, it is not a study on bridge vectors, neither on vector competence etc.

Line 525: Please modify “adult” to “female” if only females’ salivary glands were dissected.

Line 559: Consider to refer to full methodology published in former articles.

Line 637: Change “l212121” to “l212121”.

Line 646: Please move the grant information into acknowledgement/funding section.

Supplementary Fig. 1: Please explain how the pre-absorption step was performed.

Supplementary Fig. 3: Please correct the name of the protein in last row of the first paragraph of the alignment.

Reviewer #3 (Remarks to the Author):

In this manuscript the authors describe the characterization of two proteins from the *Culex quinquefasciatus* mosquito, CxD7L1 and CxD7L2, associated with the anti-hemostatic activities that are important to avoid blood clotting during insect meals. Assessing by microcalorimetry the binding of these proteins to several potential host ligands involved in hemostasis, they found, for the first time in the D7 family of proteins, that CxD7L1 binds to adenine nucleotides and nucleosides with high affinity, and experimentally suggest that this binding capacity enhances blood feeding in mammals where ADP plays a key role in platelet aggregation.

Overall, the ITC and crystallographic experiments are well conducted and the results and conclusions are well supported by the structural analysis. The results are very interesting and the paper is well written. A few minor comments are mentioned below.

Minor comments:

Line 201 "the C-terminal domain of CxD7L2 (CxD7L1-CT)" should be CxD7L2-CT ?

218 - Please add a proper citation for Phaser

Table 2 - There are missing values of Redundancy and Resolution for high and low resolution shells (should be something like 70-1.97 (2.0 – 1.97))

243 and 244 – Use consistent aminoacid nomenclatures for Arginine and Lysine (i.e R271, as used above). There are other similar inconsistencies in the text that need to be fixed.

Fig 5 legend - " CxD7L1 protein is colored in green. Inset is shown in" –sentence seems incomplete
"Electron density covering ADP" – what map is that? What sigma cutoff is shown?

637 "I212121" – should be in italic and the number one as subscript, as the usual crystallographic space group nomenclature

640 – phenix.refine has a proper citation in Acta Crystallogr D Biol Crystallogr 68, 352-67 (2012).

Reviewer: Dr. Glaucius Oliva, Institute of Physics of Sao Carlos, University of Sao Paulo, Brazil

Point to point response to reviewers:

Reviewer #1

A. Summary of the key results

The investigators screened a number of possible ligands for binding to two long form D7 proteins and found a new putative binding partner – adenine nucleotides and nucleosides, including ADP. A crystal structure is presented that reveals the binding pocket for ADP. It is noted that ADP binding may play a role in limiting hemostasis since ADP plays a role in platelet aggregation. Biochemical assays are presented showing that one D7 protein can inhibit platelet aggregation when low doses of ADP are used as an agonist.

B. Originality and significance

A number of proteins and functions have been ascribed to mosquito saliva that prevent hemostasis and facilitate blood feeding. The mechanism described here is new, but does not add much to the literature since it is well established that mosquito saliva inhibits platelet aggregation. This is a new mechanism, but it is not clear how important this mechanism is *in vivo* since only one form of D7 utilizes this mechanism, this mechanism only works with low doses of ADP, and there are many other agonists present *in vivo* that would induce platelet aggregation. D7 was able to inhibit platelet aggregation *in vitro* when ADP was used as an agonist, but it is not clear how important this is or if it would work *in vivo*. It is also not clear how important this mechanism is in individuals who are atopic and who release much higher levels of histamine and other biogenic amines, which would saturate the D7 proteins. It is possible that the impact of D7 proteins is limited to non-sensitive individuals.

We thank Reviewer #1 for her/his time and valuable comments to improve the quality and clarity of this manuscript.

In order to address the relevance of these proteins to prevent hemostasis, we performed and *in vivo* bleeding time experiment. We assessed the bleeding time in using a classical murine tail-bleeding assay. Both CxD7 proteins showed a dose dependent extension of mouse bleeding time as a result of impairment of hemostatic response to tissue injury. As expected, CxD7L1 showed a stronger more robust anti-hemostatic effect than CxD7L2. These results have been included in Results, Discussion and Methods sections (lines 407-413, lines 564-572, lines 806-813) and in Supplementary Fig. 8.

We also performed another *in vivo* experiment to quantify the ADP concentration in mouse skin before and after mosquito bites. We demonstrated a reduction of ADP levels after the bites, effects that can be attributed to the effect of CxD7L1 since the apyrase activity is not very potent in *Culex* and because its activity is prevented by performing the test with EDTA. These results have been included in Results, Discussion and Methods sections (lines 309-319, lines 503-508, and lines 787-805) and in Supplementary Fig. 6.

To demonstrate the role of CxD7 proteins in blood feeding, we generated polyclonal antibodies in rabbits against CxD7L1 and CxD7L2. These antibodies recognize specifically their correspondent recombinant proteins, and they were used in Western blot or immunolocalization studies. However, these antibodies do not block the ability of CxD7L1 and CxD7L2 to bind ADP and serotonin, respectively. These ITC experiments have been included in Discussion (lines 561-564) and in Supplementary Fig. 9. Our results indicate that performing feeding experiments on animals immunized against CxD7 is not a suitable system to study the physiological relevance of these proteins in blood feeding. Therefore, an *in vivo* evidence of these proteins role in blood feeding could only be done by significantly reducing the amount of protein in the mosquitoes, either by dsRNAi or CRISPR/Cas9 systems. We attempted to silence *CxD7L1* and *CxD7L2* transcripts by RNAi experiments, injecting dsRNA into newly emerged female adults. Silencing of D7 transcripts were very low and even less for the protein levels. We also tried 2 sets of dsRNA injections after adult emergence with similar results. It is known that salivary proteins are translated upon adult emergence and their transcripts accumulated in late pupal stages, making it very difficult to knock them down by dsRNAi (Chagas *et al* PNAS 2014; 111(19), Das *et al* BMC Genomics 2010; 11, 566). Given the technical difficulties and the time it takes to create a homozygous line by CRISPR/Cas9, we have not considered this approach for the current publication.

We do not believe that being a sensitive or non-sensitive individual would have any effect on the capacity of this proteins to interfere with their ligands and prevent host hemostasis in the bite site. Mosquito blood feeding is a fast process and it is normally completed with a few minutes. Allergic-driven mechanisms will take longer than that to be effective and by the time these antibody-mediated mechanisms are activated, the mosquito would have long be gone.

C. Data and methodology

The data presented in the manuscript appear thorough and sophisticated, although it is an *in vitro* analysis with recombinant proteins, and it is not known how important this new mechanism is *in vivo*.

Thank you.

D. Appropriate use of statistics

The statistics are fine.

Thank you.

E. Conclusions

The study is very well performed, and the conclusions are well reasoned based on the data that is presented. The impact is lessened because this is an entirely *in vitro* and biochemical analysis of a protein and putative ligand, and it is unclear how important this mechanism is *in vivo*. This is especially important since there are many redundant mechanisms in mosquito saliva that serve to prevent hemostasis and promote blood feeding.

We thank Reviewer #1 for his/her words of encouragement and finding that this work is well performed.

Several *in vivo* studies that confirmed our *in vitro* data have been included in the revised version:

- Tail bleeding assay demonstrated that CxD7, specially CxD7L1 inhibited hemostasis *in vivo* (see comment above, in section B)
- Determination of ADP levels after *Culex quinquefasciatus* mosquito bites demonstrate that mosquito bites reduced ADP levels at the bite site (see comment above, section B).

F. Suggested improvements

Negative controls would be helpful in Table 1 to show that the assay was properly calibrated.

Negative controls have been included in Tables.

It would also be helpful to determine how many D7 molecules are expectorated into skin and how many ADP molecules are expected to be present in a mosquito bite lesion in non-sensitive and hypersensitive individuals. This is critical to estimate the feasibility of D7 in scavenging enough ADP to have a physiologically relevant impact, and absolutely necessary in the absence of an *in vivo* experiment. This is lightly addressed in lines 382-386 but some experimental evidence would be helpful to determine if there really is enough D7 present in an inoculum to bind enough of this material.

We have determined CxD7 proteins concentration in salivary gland extracts of individual mosquitoes before blood feeding (n = 15) and after blood feeding (n=15). This experiment also allowed us to estimate the amount of D7 released during a bite. We also quantified D7 proteins amounts in expectorated saliva. This is addressed in Methods, Results and Discussion sections (lines 161-170, lines 491-503, and lines 690-708). We estimated the amount for D7 protein to be between 6.5-6.9 ng for CxD7L1 and 4.3-9.7 ng for CxD7L2 at the bite site. We also performed a separated experiment to determine whether the ADP levels *in vivo* are affected after a mosquito bite. We found that ADP levels are reduced approximately 40% in mouse ears bitten by *Culex* mosquitoes. This experiment indicates that CxD7 can act as an ADP scavenger *in vivo*, facilitating the intake of a blood meal.

G. References

The references are fine.

Thank you.

H. Clarity and context

The manuscript is very well written and easy to follow the text and figures. Well done.

Thank you, much appreciated!

Reviewer #2 (Remarks to the Author):

This is a complex study combining biochemical studies, crystallography and ex-vivo experiments. Authors demonstrated different ligand-binding affinities of two *Culex quinquefasciatus* D7 proteins and revealed a novel binding affinity for adenine nucleotides/nucleosides in CxD7L1 D7 protein. Introduction is comprehensive and well written, methods are written with enough details. Data acquired may prompt researchers in the field to reveal similar affinities in other *Culex* species or to enlarge current knowledge of long D7 proteins in *Anopheles* and *Aedes* mosquitoes. Results are interesting and novel, nevertheless I like to ask authors to clarify/answer some important details.

1. Did you confirm by ITC that CxD7L2 do not bind any nucleotides/nucleosides?

We thank reviewer #2 for his/her time and comments. We have addressed all of his/her comments here. Yes. CxD7L2 does not bind ADP or ATP as demonstrated by ITC. These data have been included in results (Lines 215-216) "No binding to LTB₄, ATP or ADP was detected" and also in Fig 3.

2. Could you estimate what amount of D7 proteins of interest is injected into the feeding site by *Culex quinquefasciatus* mosquito? Is the final concentration of these proteins high enough to sequester ligands presented at the bite site at 1-10 μ M concentration?

Yes, we determined the CxD7 L1 (31ng) and L2 amounts (35ng) in salivary gland extracts of individual mosquitoes before and after blood feeding (n=15 each group). This experiment also allowed us to estimate the amount of D7 released during a bite. We also determined the amount of D7 proteins in saliva. This is addressed in Results, Discussion and Method sections (lines 161-170, lines 491-503, and lines 690-708).

We also performed *in vivo* experiments to determine the ADP levels in mouse skin before and after a mosquito bite. Our results show that there is a reduction of 39.45% of ADP after mosquito bites, demonstrating that CxD7L1 can effectively scavenge ADP *in vivo*. These new experiments can be found in Results, Discussion and Method sections (lines 309-319, lines 503-508, and lines 787-805) and in Supplementary Fig. 6.

3. What is your opinion about the third D7 long form CQ_LD7_3, a protein similar to CxD7L2 (73% identity)? Did you rule out the possibility of cross-reaction between these two proteins in your immunolocalization experiment?

CQ_LD7_3 is a similar protein to CxD7L2 with a unique insert of 20 amino acids in the C-terminal domain. We performed a detailed bioinformatic analysis, qPCR and mass spectrometry of salivary gland extracts. Although CQ_LD7_3 is present in the genome, it appears that D7L3 is not transcribed in the salivary glands. As a confirmation, we did not find that protein region in the salivary glands' proteome. We believe there is a high probability that CQ_LD7_3 is a result of an error in the transcriptomic annotation of *C. quinquefasciatus* salivary glands. This discussed in lines 432-443. Besides, additions to the Results and Methods sections can be found in lines 149-160, 814-852, and in Supplementary Fig. 3.

4. Discussion is well-written, I agree with the hypotheses that this D7 protein partially substitutes less active apyrase in *Culex*. However, I would greet any supporting data that this novel ADP-binding function evolved to enhance bloodfeeding in mammals. It would be useful to compare binding activities of D7 proteins in closely related species *Culex pipiens pipiens* which is ornithophilic. I believe that such comparison would be more informative than the comparison with *C. tarsalis* which is opportunistic feeder and more distant phylogenetically form *C. quinquefasciatus*.

We would like to include *Culex pipiens pipiens* D7 in our alignment, and even purify *C. pipiens* D7L1 and test binding affinities *in vitro*. Unfortunately, *C. pipiens pipiens* transcriptome is not publicly available and we do not have a *Cx. pipiens pipiens* mosquito colony to isolate the D7 genes.

Specific comments to figures

Thanks for your comments and suggestions. We have modified the figures and text as recommended by Reviewer #2 for clarity.

1. Alignment shown in Fig. 1 includes only mosquito D7 proteins despite sand fly D7s are mentioned in the preceding text. Why you did not include, for example, *P. duboscqi* D7-related protein (PDB ID: 6MT7)?

Phlebotomus duboscqi D7-related protein (PDB ID: 6MT7) with its correspondent amino acids involved in the lipid U46619 binding has been included in the alignment.

2. Residues involved in ADP binding are not marked in the Fig. 1. On contrary, they are highlighted in blue in Suppl. Fig. 3. These highlighted amino acids present in *C. tarsalis* D7 homologs can be found also in *AeD7* (both Y conserved) and *3AnSTD7L1* (conserved Y, K and Y) in Fig 1.

Amino acids involved in ADP binding have been included in Supplementary Fig. 1, including S130.

K146 and S130 have been also highlighted in Suppl. Fig. 5., as amino acids involved in ADP-binding.

I suggest mentioning this in the text and highlighting the amino acids in in Fig. 1.

The sentence in line 297-298 have been changed to address this point: "Some, but not all of the residues involved in ADP binding were conserved in other D7 homologs".

3. In Fig. 4, result for LTB4 is not presented (in contrast to LTD4 or leukotrienes with no binding affinities). Please add the ITC curve for LTB4 into Fig. 4 (or mention in the text on line 190 that the data are not shown).

ITC curve of *CxD7L2* and LTB4 has been included in figure.

4. Could you mark all amino acids mentioned in the text following paragraph (including K146, N265, R271 and S263) in the figure 5e and point to the exact amino acid by an arrow? Missing arrows makes the figure difficult to follow.

Figure has been modified as suggested. The figure now depicts all residues within a 3.6 angstrom distance from the ADP. We also included S130, which appears to make a hydrogen bond with the oxygen of the beta phosphate of ADP (information regarding S130 has been included in the result section, Lines 270-270). GOL (corresponds to GOL-402 in the pdb file) is a glycerol molecule that is in the area of the phosphate group. R271 was not included in the figure as it is far, and it binds N265 but it does not bind ADP directly. The Figure caption has also been updated.

Minor comments or typos

Thanks again for pointing out typos in our manuscript. We have addressed all of them in the revised version of this manuscript.

Lines 17-18: Please modify "mosquitoes" to "mosquito" (or "vector" to "vectors").

It has been changed.

Line 35: Add a comma after the bracket.

A comma has been included.

Line 38: Please modify “They can also” to “It also can”.

It has been changed.

Table 1: consider to divide this table into two individual ones, each being close to the relevant ITC figure.

Table 1 has been divided into two tables, one for CxD7L1 and another for CxD7L2. Substances that do not bind have been also included in the table as suggested by Reviewer#1.

Kd values have been included in the figure to help the reader to clarify the data.
Line 166: Modify “j-f” to “f-j”.

It has been changed.

Line 200: This part is not mentioned in the methods. Please include short methodology into the section 4.4.

The following information has been included in the methodology (lines 670-676): “To determine the binding capacity of the N-terminal and C-terminal domains of CxD7L2, we independently cloned the correspondent cDNA into pCR2.1 TOPO vector (Invitrogen). CxD7L2-NT nucleotides 1 to 150 and CxD7L2-CT nucleotides 151 to 293 were amplified by PCR and subcloned into pCR2.1 After verifying the sequence identity by sequencing, the cDNA was subcloned into pET-17b plasmid between NdeI and XhoI restriction sites. Protein expression was carried out using BL21 pLysS cells (Invitrogene). N-terminal and C-terminal domains were purified as described above.”.

Line 201: Please correct “CxD7L1-CT” to “CxD7L2-CT”.

It has been changed.

Line 219: Modify “C-domains” to “C-terminal domains”.

It has been changed.

Line 260: I do not understand why you refer to Suppl. Fig. 3 here (the figure shows the sequence similarity with Culex tarsalis D7 proteins).

Agreed. The mention to Supplementary Fig. has been removed.

Line 321: Correct “Fig. 4h” to “Fig. 4i”.

It has been changed.

Line 356: Please include the concentrations of the recombinant proteins into Fig. 9.

It has been changed.

Line 370: Modify “expressed” to “localized” (protein was detected by immunolocalization).

It has been changed.

Lines 485-489: Please find more appropriate reference to support your statement. Study in

reference 46 describes platelet-activating-factor-hydrolyzing phospholipase, it is not a study on bridge vectors, neither on vector competence etc.

Reference 46 has been substituted by the following reference:

Hamer, G.L., et al. *Culex pipiens* (Diptera: Culicidae): a bridge vector of West Nile virus to humans. *J Med Entomol* 45, 125-128 (2008).

Line 525: Please modify “adult” to “female” if only females’ salivary glands were dissected.

It has been changed.

Line 559: Consider referring to full methodology published in former articles.

The following reference has been included:

Kim, I.H., Pham, V., Jablonka, W., Goodman, W.G., Ribeiro, J.M.C., and Andersen, J.F. (2017). A mosquito hemolymph odorant-binding protein family member specifically binds juvenile hormone. *J Biol Chem* 292, 15329-15339.

Line 637: Change “I212121” to “I212121”.

It has been changed to $I_{2,2,2_1}$.

Line 646: Please move the grant information into acknowledgement/funding section.

The funding sentence is part of the citation of the UCSF Chimera software. It has been removed, as it may lead to confusion.

Supplementary Fig. 1: Please explain how the pre-absorption step was performed.

It has been included in the Methods section (Protein quantification by ELISA). Lines 699-704.

Supplementary Fig. 3: Please correct the name of the protein in last row of the first paragraph of the alignment.

It has been corrected.

Reviewer #3 (Remarks to the Author):

In this manuscript the authors describe the characterization of two proteins from the *Culex quinquefasciatus* mosquito, CxD7L1 and CxD7L2, associated with the anti-hemostatic activities that are important to avoid blood clotting during insect meals. Assessing by microcalorimetry the binding of these proteins to several potential host ligands involved in hemostasis, they found, for the first time in the D7 family of proteins, that CxD7L1 binds to adenine nucleotides and nucleosides with high affinity, and experimentally suggest that this binding capacity enhances blood feeding in mammals where ADP plays a key role in platelet aggregation. Overall, the ITC and crystallographic experiments are well conducted and the results and conclusions are well supported by the structural analysis. The results are very interesting and the paper is well written. A few minor comments are mentioned below.

Minor comments:

We thank Dr. Oliva for his time reviewing this manuscript and for his comments and suggestions that indeed has improved the quality and clarity of our work presented here.

Line 201 "the C-terminal domain of CxD7L2 (CxD7L1-CT)" should be CxD7L2-CT?
It has been changed.

218 - Please add a proper citation for Phaser.

The following reference has been included.

McCoy AJ, Grosse-Kunstleve RW, Adams PD, Winn MD, Storoni LC, Read RJ. Phaser crystallographic software. *J. Appl. Crystallogr.* **40**, 658-674 (2007).

Table 2 - There are missing values of Redundancy and Resolution for high and low resolution shells (should be something like 70-1.97 (2.0 – 1.97)).

They have been included.

243 and 244 – Use consistent amino acid nomenclatures for Arginine and Lysine (i.e R271, as used above). There are other similar inconsistencies in the text that need to be fixed.
Inconsistencies have been addressed.

Fig 5 legend - " CxD7L1 protein is colored in green. Inset is shown in" –sentence seems incomplete"Electron density covering ADP" – what map is that? What sigma cutoff is shown?

Fig caption has been updated:

(c) Electron density map covering ADP. CxD7L1 protein is colored in green. (d) Inset from figure 4c is shown. Amino acid residues of CxD7L1 involved in ADP binding are colored in green (e). Stereo view of the binding pocket of the CxD7L1-ADP complex showing the $2F_o - F_c$ electron density contoured at 1σ covering the ligand. All residues within a 3.6 Å distance from the ADP are shown. Hydrogen bonds are colored in yellow. GOL-402 is a glycerol molecule that is in the area of the phosphate group. R271 was not included in the figure as it is not within 3.6 Å and it binds N265, but it does not bind ADP directly.

637 "I212121" – should be in italic and the number one as subscript, as the usual crystallographic space group nomenclature

It has been changed to *I*₂*1*₂*1*.

640 – phenix.refine has a proper citation in Acta Crystallogr D Biol Crystallogr 68, 352-67 (2012).

Phenix.refine citation has been updated.

Reviewer: Dr. Glaucius Oliva, Institute of Physics of Sao Carlos, University of Sao Paulo, Brazil

REVIEWERS' COMMENTS:

Reviewer #1 (Remarks to the Author):

Thanks. My concerns have been addressed.

Reviewer #2 (Remarks to the Author):

Thank you for accepting my suggestions. It is a pity that *Culex pipiens pipiens* was not available but I understand that it is not easy to get them.

DEPARTMENT OF HEALTH & HUMAN SERVICES

Public Health Service

National Institutes of Health
National Institute of Allergy
and Infectious Diseases
Bethesda, Maryland 20892

REVIEWERS' COMMENTS:

Reviewer #1 (Remarks to the Author):

Thanks. My concerns have been addressed.

Thank you.

Reviewer #2 (Remarks to the Author):

Thank you for accepting my suggestions. It is a pity that *Culex pipiens pipiens* was not available but I understand that it is not easy to get them.

Thank you.